



# The cryostratigraphy of the Yedoma cliff of Sobo-Sise Island (Lena Delta) reveals permafrost dynamics in the Central Laptev Sea coastal region during the last about 52 ka

Sebastian Wetterich[1]*, Alexander Kizyakov[2], Michael Fritz[1], Juliane Wolter[1], Gesine Mollenhauer[3],
Hanno Meyer[4], Matthias Fuchs[1], Aleksei Aksenov[5], Heidrun Matthes[6], Lutz Schirrmeister[1], Thomas
Opel[4,7]

[1]Permafrost Research, Alfred Wegener Institute Helmholtz Centre for Polar and Marine Research, Potsdam, Germany
[2]Cryolithology and Glaciology, Faculty of Geography, Lomonosov Moscow State University, Moscow, Russia
3Marine Geochemistry, Alfred Wegener Institute Helmholtz Centre for Polar and Marine Research, Bremerhaven, Germany
[4]Polar Terrestrial Environmental Systems, Alfred Wegener Institute Helmholtz Centre for Polar and Marine Research,
Potsdam, Germany
[5]Polar Geography, Arctic and Antarctic Research Institute, St. Petersburg, Russia
[6]Atmospheric Physics, Alfred Wegener Institute Helmholtz Centre for Polar and Marine Research, Potsdam, Germany
[7]PALICE, Alfred Wegener Institute Helmholtz Centre for Polar and Marine Research, Bremerhaven, Germany

*Correspondence to:* Sebastian Wetterich (sebastian.wetterich@awi.de)

**Abstract.** The present study examines the formation history and cryolithological properties of late Pleistocene Yedoma Ice Complex (IC) and its Holocene cover in the eastern Lena Delta on Sobo-Sise Island. The sedimentary sequence was continuously sampled in 0.5 m resolution at a vertical Yedoma cliff starting from 24.2 m above rivel level (arl). The sequence differentiates into three cryostratigraphic units; unit A: dated from ca. 52 to 28 cal ka BP; unit B: dated from ca. 28 to 15 cal ka BP; unit C: dated from ca. 7 to 0 cal ka BP. Three chronologic gaps in the record are striking. The hiatus during the interstadial MIS 3 (36–29 cal ka BP) as well as during stadial MIS 2 (20–17 cal ka BP) might be related to fluvial erosion and/or changed discharge patterns of the Lena River caused by repeated outburst floods from the glacial Lake Vitim in Southern Siberia along the Lena River valley towards the Arctic Ocean. The hiatus during the MIS 2-1 transition (15–7 cal ka BP) is a commonly observed feature in permafrost chronologies due to intense thermokarst activity of the deglacial period. The chronologic gaps of the Sobo-Sise Yedoma record are similarly found at two neighbouring Yedoma IC sites on Bykovsky Peninsula and Kurungnakh-Sise Island, and most likely of regional importance.

The three cryostratigraphic units of the Sobo-Sise Yedoma exhibit distinct signatures in properties of their clastic, organic and ice components. Higher permafrost aggradation rates of 1 m ka$^{-1}$ with higher organic matter (OM) stocks (29±15 kg C m$^{-3}$, 2.2±1.0 kg N m$^{-3}$) and mainly coarse silt are found for the interstadial MIS 3 unit A if compared to the stadial MIS 2 unit B with 0.7 m ka$^{-1}$ permafrost aggradation, lower OM stocks (14±8 kg C m$^{-3}$, 1.4±0.4 kg N m$^{-3}$ in unit B) and pronounced peaks in the coarse silt and medium sand fractions. Geochemical signatures of intrasedimental ice reflect the differences in summer evaporation and moisture regime by higher ion contents and less depleted stable $\delta^{18}$O and $\delta$D isotope ratios but lower deuterium excess (*d*) values during interstadial MIS 3 if compared to stadial MIS 2. The $\delta^{18}$O and $\delta$D composition of MIS 3





and MIS 2 ice wedges shows characteristic well-depleted values and low $d$ values, while MIS 1 ice wedges have elevated mean $d$ values between 11‰ and 15‰ and surprisingly low $\delta^{18}O$ and $\delta D$ values. Hence, the isotopic difference between late Pleistocene and Holocene ice wedges is more pronounced in $d$ than in $\delta$ values.

The present study of the permafrost exposed at the Sobo-Sise Yedoma cliff provides a comprehensive cryostratigraphic inventory, insights into permafrost aggradation and degradation over the last about 52 thousand years, and their climatic and morphodynamic controls on the regional scale of the Central Laptev Sea coastal region in NE Siberia.

## 1 Introduction

During sea level low stands of the last glacial period, vast areas of the East Siberian arctic shelves were exposed and formed the unglaciated Beringia Land Bridge between the Eurasian Scandinavian and Barents-Kara ice sheets, and the North
American Laurentide ice sheet (Hopkins, 1959). Beringia hosted a unique cold-adapted ecosystem with no analogue in modern times − the tundra-steppe that maintained the late Pleistocene Mammoth fauna (Hopkins, 1982). Beringian environments were characterised by permafrost formation in widespread ice-wedge polygonal networks (Sher, 1997). The ice-wedge polygons grew syngenetically, i.e. contemporaneously with deposition of ice-rich clastic and organic material. Generally, ice wedge formation takes place after thermal contraction cracking of the frozen ground in wintertime,
consecutive infill of the cracks by meltwater in spring and immediate refreezing and expansion (Leffingwell, 1915). The vertical ice veins formed by annual repetition of this mechanism widen the wedge ice, which in addition grows upwards with ongoing deposition. For about 70 thousand years (kiloannum, ka) during the Marine Isotope Stages (MIS) 4, 3 and 2, the Beringian tundra-steppe environment accumulated up to 50 m thick ice-wedge polygon sequences that are named in Russian stratigraphy as Yedoma Ice Complex (IC; Tumskoy, 2012). Yedoma IC was first described by Soloviev (1959) in Central
Yakutia, while Katasonov (1954/2009) and Romanovskiy (1959) undertook first cryostratigraphic research of Yedoma IC in the East Siberian lowlands. Yedoma IC formation is characterised by cryogenic cyclicity (Popov, 1953; Vasil'chuk, 2013) that is expressed in distinct horizons, which formed in relation of the deposition rate to interannual variations in active-layer depth and result in freezing events that built a respective uppermost portion of perennially frozen ground (Wetterich et al., 2014). Diagnostic for Yedoma IC are the presence of syngenetic ice wedges, the oversaturation of the sediment with pore ice
and segregated ice (excess ice) forming lenticular and reticulate cryostructures within mainly fine-grained deposits (for an overview see Schirrmeister et al., 2013).

After deglaciation at the end of the last glacial maximum (LGM), sea level rise and subsequent inundation of the East Siberian shelves flooded large parts of Beringia whose remaining areas were further affected by intense permafrost degradation caused by deglacial warming. The LGM distribution of the Yedoma domain included now submerged shelf
areas (1.9 million km$^2$) and the modern maximum extent of Yedoma deposits on land (1.4 million km$^2$). This amounts to an area of about 3.3 million km$^2$ where Yedoma IC formation potentially took place (Strauss et al., 2017 and references therein). In the modern terrestrial Yedoma domain, up to 70 % of the area are affected by thermokarst (Strauss et al., 2013),



which is the degradation of ice-rich permafrost due to thaw, ground subsidence and erosion. In Eurasia, Yedoma IC deposits are most widespread in the accumulative lowlands of northern and Central Yakutia, less on Taimyr and Chukotka
(Romanovskiy, 1993; Kunitsky, 2007; Konishchev, 2011; Grosse et al., 2013). The most widespread distribution of IC is characteristic of lowland plains at altitudes of less than 100 m above sea level (asl). Inland Yedoma IC has, however, also been found in e.g. the Yana uplands exposed in the Batagay megaslump (Kunitsky et al., 2013; Murton et al., 2017, under review; Opel et al., 2019). In North America, deposits similar to the Siberian Yedoma IC occur in lower parts of the Arctic Foothills, in the northern part of Seward Peninsula, in interior Alaska and in the Yukon Territory (Péwé, 1955; Sanborn et
al., 2006; Kanevskiy et al., 2011).

Ongoing research of Yedoma IC employs its clastic component to unravel material sources, transformation, transportation and sedimentation processes by applying mineralogy, grain-size analysis and end-member modelling approaches (e.g., Murton et al., 2015; Schirrmeister et al., 2011a; Schirrmeister et al., 2020; Strauss et al., 2012) to reveal the depositional history. The organic component of Yedoma IC bears information on the Beringian environment and its variability over time
in floral and faunal fossil records (e.g., Sher et al., 2005). The organic matter (OM) preserved in Yedoma IC was studied for its carbon stocks (e.g., Strauss et al., 2013; Zimov et al., 2006) and carbon vulnerability to estimate its degradability upon thaw (e.g., Stapel et al., 2016; Strauss et al., 2015). To obtain chronologies of Yedoma IC formation, radiocarbon ($^{14}$C) dating of organic remains is commonly applied (e.g., Schirrmeister et al., 2002a; Wetterich et al., 2014). The ice component comprises intrasedimental ice and wedge ice, which together constitute the major share of up to about 80 percent per volume
(vol%) of the Yedoma IC (Strauss et al., 2013). Cryostructures of intrasedimental ice as well as its stable water isotope composition host information of past freezing conditions (e.g., Dereviagin et al., 2013; Schwamborn et al., 2006). The stable isotope composition of wedge ice is more often used in palaeoclimate studies and serves as proxy of winter climate conditions and moisture sources because the wedge ice derives mainly from winter precipitation (e.g. Lachenbruch, 1962; Opel et al., 2018).

The present study in the eastern Lena Delta fills a geographic gap in the extensive Yedoma IC studies in the Laptev Sea coastal region, which were executed during the last about two decades in joint Russian-German research (Khazin et al., 2019; Meyer et al., 2002a; 2002b; Schirrmeister et al., 2002a; 2011a; 2017; Sher et al., 2005; Strauss et al., 2013, 2015; Tumskoy, 2012; Wetterich et al., 2005; 2008a; 2011, 2014). Our study seeks (1) to capture and to characterise the entire cryostratigraphic inventory of frozen deposits and ground ice of the Sobo-Sise Yedoma cliff, (2) to estimate permafrost
aggradation and degradation timing, extent and processes on Sobo-Sise Island in context of widespread Yedoma IC occurence in the Central Laptev Sea coastal region, (3) to decipher regional winter climate conditions and moisture sources and (4) to disentangle the relation of permafrost dynamics controlled by large-scale climate variability and regional to local geomorphologic conditions and processes.


## 2 Study area

The Lena Delta stretches at the shore of the Laptev Sea between about 72°N to 74°N and 123° to 130°E (Fig. 1), and is the largest Arctic river delta (Walker et al., 1998). The terrestrial surface of the delta differentiates into three geomorphological units (or terraces; Grigoriev, 1993). The Holocene-aged first terrace comprises the northeastern and the south-southwestern parts of the delta and is mainly covered by wet polygonal tundra and thermokarst basins (Morgenstern et al., 2008). The second terrace in the northwestern part of the delta is composed of fluvial dry sands dated to MIS 3-2 (Schirrmeister et al.,

2011b), and is characterised by numerous NNW-SSE-oriented lake basins and less expressed polygonal surface morphology (Morgenstern et al., 2008). The third terrace of MIS 4-2 age occurs in the southern part of the delta. The study area on Sobo-Sise Island in the southeastern part of the delta belongs to the third geomorphologic terrace that is shaped by remnants of late Pleistocene Yedoma IC and its degradation features. According to a landform classification of Sobo-Sise by Fuchs et al. (2018), 43 % of the land surface are occupied by Yedoma uplands and partly degraded Yedoma slopes, 43 % are thermokarst

basins and 14 % are lakes. The studied Yedoma cliff in the northern part of Sobo-Sise facing the Sardakhskaya Channel is remarkable for its rapid shoreline retreat of up to 22.3 m yr$^{-1}$ (based on remote sensing time series 1965-2018; Fuchs et al., under review) with a mean annual retreat rate of 9.1 m yr$^{-1}$ over the observation period.  The resulting annual OM release into the Lena River amounts to at least 5.2 x 10$^6$ kg organic C and 0.4 x 10$^6$ kg N per year for the period 2015-2018 (Fuchs et al., under review). Furthermore, the Sobo-Sise Yedoma undergoes elevation changes due to thaw subsidence between −2 cm

yr$^{-1}$ (based on Sentinel-1 InSAR, 2017, for the entire Sobo-Sise Island; Chen et al., 2018) and −3.4 cm yr$^{-1}$ (based on on-site rLiDAR at the studied Yedoma cliff, Günther et al., 2018), indicating ongoing permafrost degradation. Thus, the Sobo-Sise Yedoma represents a typical OM source in land-to-ocean pathways. It is characterised by substantial OM stocks and fast permafrost degradation (Fuchs et al., under review) accelerated by Arctic warming (Fritz et al., 2017). The Yedoma cliff rises to about 27.7 m above river level (arl; Fuchs et al., under review) and stretches about 1.66 km in NW to SE direction

(Fig. 1). It is likely that the Yedoma IC extends up to about 12 m below the river level as deduced by Fuchs et al. (under review) from near-shore bathymetry in front of the cliff.

The modern climate of the Lena Delta as recorded by ongoing monitoring on Samoylov Island in the central delta reveals a mean annual air temperature of −12.3 °C (1998–2017; Boike et al., 2019). Mean monthly air temperatures reach 9.5 °C in July and −32.7 °C in February. The average annual rainfall amounts to 169 mm and the average annual winter snow cover to

0.3 m (2002–2017; Boike et al., 2019). Between 2006 and 2017, permafrost has warmed by 1.3 °C at the zero annual amplitude depth of 20.75 m (Boike et al., 2019) while the permafrost maximum depth in the region reaches 500–600 m (Grigoriev, 1993). Unfrozen underground (talik) is, however, assumed below the main channels of the Lena Delta and below thermokarst lakes exceeding 2 m water depth.

The modern vegetation of Sobo-Sise Island is mainly characterised by dwarf shrub - moss - tussock tundra communities

occupying varying habitats of the land surface of Yedoma uplands, pingos, floodplains, and thermokarst basins. Common



species belong to the genera *Salix*, *Dryas*, *Saxifraga*, *Polygonum*, *Carex*, *Poa*, *Trisetum*, *Equisetum*, *Luzula* and unspecified mosses and lichen according to Raschke and Savelieva (2017).

## 3 Material and methods

### 3.1 Fieldwork

We sampled profiles at different positions of the Yedoma cliff including three vertical sediment profiles and six horizontal ice-wedge profiles to cover the entire exposed permafrost inventory (Fig. 2). The sequence was cryolithologically described according to French and Shur (2010) and frozen samples were obtained using hammer and axe at 0.5-m resolution via rope descending. Vertical overlaps of the three profiles of sedimentary polygon fillings of the exposure ensured complete sampling coverage of the cliff (Fig. 2). Sample positions of profiles SOB18-01 and SOB18-03 were measured as depths in

meters below surface (m bs) and transferred to heights from the measured height of 24.2 m arl at the cliff edge above the profiles. Sampling positions of the lowermost profile SOB18-06 were directly measured as heights in m arl. In total, 61 sediment samples were taken (Wetterich et al., 2019). The gravimetric ice/water content was measured in the field as difference between wet and dry weights after careful drying on a field oven and is given as weight percent (wt%).

Three horizontal ice-wedge profiles at the Yedoma IC cliff were sampled in 2018 at different height levels (Fig. 2).

Additional three ice-wedge profiles sampled in 2014 were analysed and included in the present study. Ice-wedge profile SOB18-02 was taken at 19.7 m arl on rope by ice screw at 15-cm resolution between the sediment profiles SOB18-01 and SOB18-03 in the uppermost part of the cliff (Fig. 3 b). The ice wedge SOB18-09 derives from the central western part of the Yedoma and was sampled at 2 m arl at beach level (Fig. 3 f). The ice-wedge profiles SOB18-08 and SOB14-IW4 were sampled at the upper western slope of the Yedoma towards the western alas basin at 9.4 m arl and 9 m arl, respectively (Fig.

3 e, h). Ice-wedge profiles in the lower part of the cliff were obtained at 2.5 m arl (eastern slope, SOB14-IW3) and at 2 m arl (western slope, SOB14-IW5) (Fig. 3 g, i). Except for profile SOB18-02, all ice-wedge profiles were taken by chainsaw either in blocks for subsampling in the cold lab at 1.5- to 2-cm resolution or in slices in the field at varying resolution of 4 cm (SOB18-08, SOB18-09) and of 15 cm (SOB14-IW3) depending on fieldwork logistics (Wetterich et al., 2019).

### 3.2 Laboratory analyses

#### 3.2.1 Sediment and organic-matter analyses

Upon arrival in the laboratory, the sediment samples were freeze-dried (Zirbus Subliminator 3–4–5), manually homogenised and split for further analyses. The grain-size distribution (GSD) was measured using a laser diffraction particle analyser (Malvern Mastersizer 3000). GSD was calculated with the internal software of the laser diffraction particle analyser and further analysed using GRADISTAT 8.0 (Blott and Pye, 2001) for sand-silt-clay distribution, arithmetic mean in μm and





sorting in phi (φ). Further details of GSD sample preparation and laboratory procedures are given in Schirrmeister et al. (2020). The sample SOB18-03-03 was not included in the data interpretation because of analytical artefacts.

To attempt an unmixing of the measured grain-size distributions into underlying characteristic grain-size sub-populations associated with specific sedimentological deposition and transformation processes, a robust endmember modelling approach (EMMA) following Dietze and Dietze (2019) was applied to a total of 56 GSD matrices from profiles SOB18-01, SOB18-03

and SOB18-06 representing the Yedoma IC and excluding the uppermost Holocene cover. EMMA is a type of eigenspace analysis with the capacity to transform the resulting endmember components so that the loadings of the endmembers can be interpreted as grain-size distributions (see details in Dietze et al., 2012). Each sample used in the analysis is then represented as a linear combination of the identified endmembers, where the scores provide a quantitative estimate of how much an endmember contributes to a sample. The used R package EMMAGeo (Dietze and Dietze, 2019) additionally allows the

identification of robust endmembers (rEM) using a multiple-parameter approach where rEMs are those that occur independently from model parameters. A Monte-Carlo approach is then used for assessing the uncertainties associated with the scores computed for each sample. Overall class-wise explained variance is 63%, with lowest $R^2$ occurring for the very fine and very coarse classes (Fig. S1). Overall, sample-wise explained variance is 91%, with only one sample with an explained variance below 80% (Fig. S1).

Mass-specific magnetic susceptibility (MS) as a proxy for sediment content of magnetisable minerals was measured using a Bartington Instruments MS2 equipped with an MS2B sensor. MS data are expressed in SI units ($10^{-8}$ m$^3$ kg$^{-1}$). Total nitrogen (TN) and total organic carbon (TOC) contents of the samples were measured with elemental analysers (ElementarVario EL III for TN and ElementarVario MAX C for TOC; analytical accuracy ± 0.1 wt%). The ratio of TOC and TN is referred to as C/N. Stable carbon ($\delta^{13}$C) and nitrogen ($\delta^{15}$N) isotope analysis was undertaken using a Thermo Scientific

Delta V Advantage Isotope Ratio MS equipped with a Flash 2000 Organic Elemental Analyser using helium as a carrier gas. Values are given as per mil (‰) difference from the Vienna Pee Dee Belemnite (VPDB) standard for $\delta^{13}$C and from nitrogen in ambient air (AIR) for $\delta^{15}$N. The accuracy was better than ± 0.15 ‰ for $\delta^{13}$C and ± 0.2 ‰ for $\delta^{15}$N. Further details on OM analyses are given in Davidson et al. (2018). In total, 61 sediment samples were analysed for the parameters described above (Table 1).

**3.2.2 Ground-ice analyses**

Supernatant water of thawed sediment from 53 samples (Table 1) was decanted in the field. Hydrochemical characterisation included electrical conductivity (EC), major anions and cations as well as dissolved organic carbon (DOC) concentration. For major ion analyses 17 samples were filtered through 0.45 μm CA syringe filters and filled into sample bottles. HNO$_3$ (65% Suprapur) was added to cation samples for conservation. The cation content was analysed by inductively coupled

plasma-optical emission spectrometry (ICP-OES, Perkin-Elmer Optima 3000 XL), while the anion content was determined by ion chromatography (IC, Dionex DX-320). Ion concentrations are given in mg L$^{-1}$. For DOC concentration analyses,



samples were filtered through pre-rinsed 0.7 μm GF/F glass fiber filters attached to a rubber-free syringe. The liquids were filled in clear glass vials with screw caps and PTFE septum, and acidified with HCl (30% Suprapur). All samples were stored cool and dark. DOC concentrations (mg L$^{-1}$) were measured in 29 samples with a high-temperature (680 °C)

combustion TOC analyzer (Shimadzu TOC-VCPH).

The oxygen (δ$^{18}$O) and hydrogen (δD) stable isotope compositions of melted samples of intrasedimental ice and of wedge ice were measured using a Finnigan MAT Delta-S mass spectrometer (1σ < 0.1‰ for δ$^{18}$O, 1σ < 0.8‰ for δD; Meyer et al., 2000). Values are given as per mil (‰) difference from the Vienna Standard Mean Ocean Water (VSMOW) standard. The deuterium excess (*d*) is calculated following Dansgaard (1964) in Eq. (1):


$$d = \delta D - 8*\delta^{18}O \hspace{4cm} (1)$$

In total, 511 samples of wedge ice were analysed for their stable water isotope composition. We excluded marginal samples from further interpretation at the interface of wedge ice to sediment, which showed indications of isotopic alteration.

Furthermore, the sampled ice-wedge profile SOB14-IW4 captured a polygon junction and included ice of two neighbouring ice wedges. We considered only a full profile of one wedge cut perpendicular to its growth direction and neglected the remaining samples of the second, not fully captured ice wedge with an oblique cut. In total, 412 wedge ice samples were interpreted from six analysed ice-wedge profiles (Table 2).

### 3.3    Radiocarbon dating and age modelling

The geochronology along each profile was established on the basis of 32 Accelerator Mass Spectrometry (AMS) radiocarbon dates from 31 selected sediment samples (Table 3) using a Mini Carbon Dating System (MICADAS)    at Alfred Wegener Institute Helmholtz Centre for Polar and Marine Research (AWI). Further details on laboratory procedures and sample pretreatment are given in Opel et al. (2019). Notably, depending on size, samples were analysed as graphite or gas targets; the small sample size of some gas targets causes a reduced age range for which reliable radiocarbon ages can be obtained (Table

3). The dated material was obtained by hand picking of terrestrial plant remains from freeze-dried samples. In a first batch, 26 samples were chosen every 1-1.5 m, representing sediment horizons and their boundaries. On the basis of the obtained dates, five additional samples were chosen on either side of suspected hiatus to verify and delimit them more reliably. All radiocarbon dates were calibrated using the IntCal13 calibration dataset (Reimer et al., 2013). Ages are given as calibrated years before present (cal BP).

A total of 27 radiocarbon dates were used in the Bayesian age-height model, which was established using the package rbacon 2.3 (2.3.9.1) (Blaauw and Christen, 2019) in R version 3.6.1 (R Core Team, 2019). Four ages were not used for the age-height modelling: sample SOB18-03-17 (15 294 ± 67 BP) is most likely redeposited (see section 5.1.1 for discussion); two of the dates have unspecified infinite ages of >42 600 BP (SOB18-06-18 and SOB18-06-33), and one further age of 47 021 ±
646 BP (SOB18-06-35) was beyond calibration range (Fig. 4). All three ages do, however, support the age-height relation as they are of similar ballpark age.


Each profile was modelled individually (Fig. S2). Section thickness was adjusted in relation to sampling frequency along each profile to balance model performance (Blaauw and Christen, 2019). Sediment profile SOB18-01 was modelled using twelve $^{14}$C dates, a section thickness of 10 cm and a hiatus at 1.75 m bs (22.45 m arl) and 3.25 m bs (20.95 m arl). Profile SOB-03 was modelled using seven $^{14}$C dates, a section thickness of 30 cm and a hiatus at 7.25 m bs (16.95 m arl). This


model was extrapolated 0.5 m beyond the uppermost (youngest) dated sample to cover the entire profile. Profile SOB18-06 was modelled using eight $^{14}$C dates, a section thickness of 40 cm and was extrapolated 2.3 m beyond the lowermost radiocarbon age used in the model to cover the lowermost samples of this profile. SOB18-06 represents the oldest deposits of the dataset and infringes on the age limit of radiocarbon dating (Fig. S2). The median of the modelled probability distribution was used to assign an age to each cm along the profile.


Additional age information was obtained from one mammoth tusk found at beach level below the sediment profile SOB18-01 (Fig. 3 c) and from host sediments of ice wedges SOB18-08 and SOB18-09 (Table 3). Floral and faunal remains from inside wedge ice were dated where available (Table 3). In total, age information for five ice wedges was obtained from 22 radiocarbon dates. Of those, 19 samples were dated at the MICADAS facility mentioned above and three samples were dated at the CologneAMS (University of Cologne, Germany) whose laboratory procedures are given in detail in Rethemeyer et al.


(2013).

## 4 Results

### 4.1 Chronostratigraphy

The three sediment profiles were sampled in close proximity to cover the entire exposed permafrost sequence at 0.5 m sampling resolution in spatial context. The overlap in sampling heights was applied to account for possible relief diversity


during permafrost aggradation (Fig. 2). Profile SOB18-01 covers the uppermost part of the exposure between 24.2 to 15.5 m arl, dated from 2 440 to 27 540 cal BP. The adjacent sedimentary polygon filling was sampled in profile SOB18-03 between 18.8 and 10.2 m arl, dated from 25 680 to 40 840 cal BP. The lowermost profile SOB18-06, sampled about 120 m east of the SOB18-01 between 13.4 and 0.8 m arl, shows ages from 41 420 to >50 000 cal BP; Table 3). The overlap in sampling positions of the three profiles (Fig. 4) and the modelled age-height relation allows for deducing a stacked record that


differentiates into three chronostratigraphic units:

    unit A: MIS 3, Yedoma IC (52 to 28 cal ka BP),
    unit B: MIS 2, Yedoma IC (28 to 15 cal ka BP),
    unit C: MIS 1, Holocene cover (7 to 0 cal ka BP).




The stacked sequence is not continuous and shows three temporal gaps in the record, which are related to changes in the depositional and/or erosional regimes. Those are discussed in detail in section 5.3. One hiatus is obvious within unit A (in profile SOB18-03) between about 36 and 29 cal ka BP, one hiatus within unit B (in profile SOB18-01) between 20 and 17 cal ka BP and one hiatus between units B and C (in profile SOB18-01) between about 15 and 7 cal ka BP (Fig. 4).

Age inversions are often observed in permafrost chronologies given the effect of cryogenic processes such as cryoturbation within the uppermost thawed active layer before the material enters the perennially frozen state (Bockheim et al., 2007), and the high vulnerability of ice-rich permafrost to thaw and erosion (Grosse et al., 2011; Günther et al., 2015). An obvious example for the latter is seen in the Sobo-Sise record where sample SOB18-03-17 has an age of 15 294 ± 67 BP (18 570 cal BP) while the entire profile SOB18-03 dates from 25 680 to 40 840 cal BP (Table 3). We assume that this age of 18 570 cal

BP most likely represents a contamination from thawed sediment, which was redeposited downwards along the cliff and represents a mixed age of Holocene and older OM. To validate this assumption, additional plant material from sample SOB18-03-17 was picked and dated to 40 840 cal BP, in line with the lower age limit of profile SOB18-03.

Three ice-wedge profiles of unit A were dated. Seven $^{14}$C dates from ice wedge SOB18-09 range from 48 660 to 36 970 cal BP. We found one infinite age of >48 500 BP from the host deposit at the same height level as the sampling transect. Ice

wedge SOB14-IW3 shows ages of 43 270 and of 30 930 cal BP, and two infinite ages of >31 000 BP each. Organic material from ice wedge SOB18-08-I yielded a $^{14}$C age of 49 610 cal BP.

The SOB18-02-I wedge ice of unit B was dated by two ages of 25 350 and of 23 470 cal BP. The host deposits at the same height level as the sampling profile at about 18 to 20 m arl show an age range from 23 170 to 21 940 cal BP that is in general agreement with the assumed SOB18-02-I formation time. The only direct age information from unit C wedge ice is available

for profile SOB14-IW4 with eight ages spanning from 2 290 cal BP to modern (Table 3). Indirect age information is available for SOB18-08-II whose host deposits at the same height level as the sampling transect were dated to 4 480 cal BP (Table 3) implying a middle to late Holocene formation of this ice wedge.

Relocated material might also enter wedge ice when wintertime frost cracks are filled with snowmelt transporting OM and preserving it in vertical ice veins (Opel et al., 2018). This might be the case for the age determination of 49 610 cal BP in IW

SOB18-08-I that is, however attributed to unit A of MIS 3 age by its isotopic composition.

## 4.2 Cryostratigraphy

Each cryostratigraphic unit is characterised by its specific clastic, organic and ice composition. Those were captured by field observations (Wetterich et al., 2019) and analytical data that are described in detail below and summarised in Fig. 5, Fig. 6 and Table 1. Representation of our analytical results is based on the modelled age-height relation for each profile and their

stacking by age (Fig. S2).

Stable water isotope records and age information of six horizontal ice wedge profiles sampled at the Sobo-Sise Yedoma cliff are attributed to the cryostratigraphic units A, B and C by position (Fig. 7), isotopic composition (Table 2) and age (Table 3). Two of the ice-wedge profiles (SOB18-08 and SOB18-02) exhibited stable isotope compositions pointing to different stages





of ice-wedge formation, which formed under contrasting climatic conditions and therefore represent different time periods.
These differentiations are explained in detail below. Spatial dimensions and other field observations of the wedge ice are summarised in Wetterich et al. (2019). Note that marginal ice wedge samples that underwent isotopic exchange with the host sediment (Meyer et al., 2002a) are excluded from summary statistics given in Table 2 although being shown in Fig. 7 as grey symbols for completeness of the raw data.

### 4.2.1 Unit A (MIS 3, Yedoma IC, 52 to 28 cal ka BP)

The frozen deposits of unit A are represented by the entire sediment profile SOB18-06 and most of sediment profile SOB18-03 (except for its uppermost two samples that belong to unit B; Fig. 4). Unit A is evenly composed of grey poorly sorted sandy silt (mean grain-size of $45 \pm 12$ μm) with a pronounced peak in the coarse silt fraction and a minor peak in the middle sand fraction (Fig. 8 c).

End-member modelling of grain-size distributions revealed four robust endmembers (rEMs) for the Yedoma IC units A and
B. In unit A, the coarse silt rEM2 (primary mode at 31 μm) dominates while the fine silts rEM1 (primary mode at 6 μm) and fine sand rEM 3 (primary mode at 76 μm) occur less frequently (Fig. 5). The mean magnetic susceptibility of unit A is $40 \pm 9$ SI.

The OM of unit A deposits consists of numerous twigs and grass remains, black and brownish spots (0.1–0.5 cm in diameter), single peaty lenses (15–20 cm in diameter), and peat layers (10–20 cm up to 130 cm thick). The mean TOC and
TN values are $4.5 \pm 2.5$ wt% and $0.3 \pm 0.1$ wt%, respectively; C/N is $12.9 \pm 2.5$. The corresponding mean stable isotope values are $-27.3 \pm 0.9$ ‰ for $\delta^{13}C$ and $2.2 \pm 0.6$ ‰ for $\delta^{15}N$, respectively.

The organic component represented as DOC in intrasedimental ice of unit A shows large variations between 161 and 754 mg $L^{-1}$ at a mean value of 367 mg $L^{-1}$.

The intrasedimental ice of unit A amounts to a gravimetric ice content of $49 \pm 10$ wt%, manifested in lenticular
cryostructures and reticulate and wavy, sometimes structureless cryostructures between the ice layers. The stable water isotope composition of the intrasedimental ice shows a mean value of $-23.9 \pm 2.0$ ‰ for $\delta^{18}O$ and $-190 \pm 15$ ‰ for $\delta D$ (Fig. 9 a). The $d$ values also vary considerably between $-6$ ‰ and 12 ‰ (mean of 1.5 ‰).

The electrical conductivity of the intrasedimental ice of unit A shows a large variability between about 730 and 5780 μS $cm^{-1}$ (mean $2245 \pm 1570$ μS $cm^{-1}$) and is thus higher than in units B and C. The cations Ca and Mg dominate the hydrochemical
composition suggesting also substantial concentrations of $HCO_3$ that was, however, not measured due to limited sample amount. Na and Cl concentrations also reach higher values and dominate the peak in ion content around 44–43 cal ka BP (Fig. 6). At this peak also increased Fe concentrations are notable.

The ice wedge SOB18-09 of unit A reveals an isotopic composition with mean values of $-29.7$ ‰ in $\delta^{18}O$, $-232$ ‰ in $\delta D$ and $d$ of 5.2 ‰. SOB14-IW3 shows identical mean values of $-29.7$ ‰ in $\delta^{18}O$ and $-231$ ‰ in $\delta D$, but a higher $d$ of 7.2 ‰.



The eastern part of profile SOB18-08 (differentiated as SOB18-08-I) is characterised by more depleted values in δ18O (by 4 ‰), δD (by 40 ‰) and *d* (by 4‰) if compared to the main part of the profile (SOB18-08-II), which is attributed to unit C. Mean values of SOB18-08-I are –29.6 ‰ in δ18O,  –230 ‰ in δD and 6.8 ‰ in *d*, close to the respective values of the other two ice wedge profiles of unit A.

In summary, the ice wedges of unit A show most depleted mean values down to –29.9 ‰ in δ18O (range from –31.4 ‰ to –26.9 ‰) and –232 ‰ in δD (range from –244 ‰ to –213 ‰). They plot mainly below the GMWL and show low *d* between 5.2 to 7.4 ‰ in comparison to IWs of units B and C. The slopes in co-isotopic plots of ice wedge data from unit A vary between 7.2 and 8.3 (Fig. 9 c).

**4.2.2 Unit B (MIS 2, Yedoma IC, 28 to 15 cal ka BP)**

Unit B comprises the uppermost two samples of sediment profile SOB18-03 and most of sediment profile SOB18-01 (except of its uppermost four samples that belong to unit C; Fig. 4). Unit B is composed of brownish grey poorly sorted sandy silt (mean grain-size 113 ± 64 μm) and occasional sand lenses resulting in a bi-modal GSD and pronounced peaks in the coarse silt and medium sand fractions (Fig. 8 b).

Generally coarser grain-size distributions than in unit A are characteristic for unit B and supported by the EMMA results. The middle sand rEM4 (primary mode at 310 μm) is present in the lower part of unit B while fine silt rEM1 and coarse silt rEM2 dominate the upper part of unit B (Fig. 5). The fine sand rEM3 is less frequent in unit B compared to unit A.

The magnetic susceptibility of unit B has a mean of 53 ± 9 SI. OM is present as single twig remains (2-4 mm in diameter), dark brown spots, finely dispersed organic remains, and peaty lenses (5 to 25 cm in diameter). The mammoth tusk found at beach level below the profile SOB18-01 most likely originates from unit B deposits belonging to the faunal component of OM. It was radiocarbon-dated to 16 480 cal BP and thus fits into the age range of unit B. This finding fits well into the fossil record of the late Pleistocene mammoth fauna in the region (Kuznetsova et al., 2019). The OM content of unit B is lower compared to that of unit A with mean values of 2.1 ± 1.3 wt% for TOC and 0.2 ± 0.1 wt% for TN, resulting in mean C/N of 10.5 ± 2.4.

The OM isotopic composition exhibits lower mean values than in unit A of –26.1 ± 0.6 ‰ for δ13C and 1.9 ± 1.0 ‰ for δ15N. The DOC content of intrasedimental ice of unit B is generally lower compared to those of unit A with values from 85 to 589 mg L⁻¹ (mean of 212 mg L⁻¹).

The ice content of unit B is the lowest of all units with 43 ± 10 wt%. Prevailing cryostructures are lenticular (1–5 cm thick ice layers in 1–20 cm distance), and reticulate (1–2 mm thick ice lenses 4–12 mm long) or wavy parallel (1 mm thick ice lenses 4–10 mm long) between the ice layers. If compared to unit A, the intrasedimental ice of unit B shows similar mean values and comparable ranges in δ18O of –26.2 ± 2.2 ‰, and in δD of –200 ± 16 ‰ (Fig. 9 a). The mean *d* value of about 10 ‰ is much higher than in unit A, ranging from 3 ‰ to 15 ‰.





The hydrochemical composition of intrasedimental ice of unit B shows an upward decreasing trend in ion content with electrical conductivity ranging from about 3180 to 1130 µS cm$^{-1}$ (mean: 1810 µS cm$^{-1}$) ( Fig. 6). Ca and Mg cations dominate the cation composition while Cl concentrations decrease upwards.

The only IW record of unit B was obtained in the eastern part of profile SOB18-02. Like SOB18-08-I, the isotopic
composition of SOB18-02-I differs from the western part of its profile (SOB18-02-II attributed to unit C) by more depleted isotopic mean values and a lower $d$; –28.8 ± 0.5 ‰ in δ$^{18}$O, –225 ± 5 ‰ in δD and $d$ of 5.8 ± 0.9 ‰. The values plot below the GMWL and the co-isotopic plot shows a slope of 9.4 (Fig. 9 c).

### 4.2.3 Unit C (MIS 1, Holocene cover, 7 to 0 cal ka BP)

The uppermost four samples of sediment profile SOB18-01 represent the cryostratigraphic unit C including the uppermost
seasonally thawed active layer (of 0.2 m on 20 July 2018 at the sampling site) that consists of modern vegetation and peat (Fig. 4). Below the active layer, grey poorly sorted sandy silt is present. Its grain-size distribution is bi-modal with peaks in the coarse silt and medium sand fractions (mean grain-size of 66 ± 13 µm; Fig. 5, Fig. 8 a). The mean MS is the lowest of all units with 32 ± 23 SI, which corresponds to the highest OM content (present in numerous peaty lenses, 2 to 25 cm in diameter) with mean TOC of 11.3 ± 9.9 wt% and mean TN of 0.6 ± 0.3 wt%. The C/N is highest for all units with a mean
value of 18.5 ± 8. The OM stable isotope composition exhibits the most depleted mean value of –28.0 ± 02 ‰ for δ$^{13}$C of all units and a mean value of 2.1 ± 0.7 ‰ for δ$^{15}$N. From unit C only one measurement of DOC of intrasedimental ice is available showing the lowest value of the entire DOC data with 34 mg L$^{-1}$.

The protective layer underlying the thawed active layer (Kanevskiy et al., 2017) is characterised by a high ice content of 80 wt% representing ice segregation towards the freezing front during the annual freeze-thaw cycles. The cryostructures are
reticulate with 2–4 mm thick and 4–10 mm long ice lenses. Below, lenticular (up to 10 mm thick ice layers in 2–8 cm distance) and reticulate (1–4 mm thick ice lenses, 6–12 mm long) cryostructures are present. The stable water isotope composition of intrasedimental ice is less depleted than in units A and B with mean values of –20.7 ± 0.3 ‰ in δ$^{18}$O and – 151 ± 2 ‰ in δD, and shows only low variation (Fig. 9 a) likely due to the low sample number (n = 3). The $d$ value is highest for all units with about 15 ‰.

The hydrochemical composition in intrasedimental ice of unit C was characterised in only one sample, which shows a very low electrical conductivity of 36 µS cm$^{-1}$, and major ion concentrations of less than 2 mg L$^{-1}$ except for Ca and Fe.

One complete IW profile (SOB14-IW4) and two profile parts (SOB18-08-II and SOB18-02-II) belong to the cryostratigraphic unit C of Holocene cover deposits. We furthermore consider the Holocene ice wedge profile SOB14-IW5 from the lowermost part of the Yedoma slope that might represent a former thermokarst basin (alas) level overlying Yedoma
IC. The isotopic composition of SOB14-IW4 shows mean values of –27.8 ‰ in δ$^{18}$O, –207 ‰ in δD and 14.9 ‰ in $d$. Less depleted values in δ$^{18}$O of –27.3 ‰, –26.2 ‰ and –25.2 ‰ are found in profiles SOB14-IW5, SOB18-08-II and SOB18-02-II, respectively. The according δD mean values from these three profile are –204 ‰, –196 ‰ and –190 ‰ with $d$ values of



13.8 ‰, 13.8 ‰ and 11.2 ‰. All wedge ice records of unit C are clearly distinguished from those of units A and B by less depleted $\delta^{18}O$ and $\delta D$ and $d$ values well above 10 ‰ (Table 2). They plot predominantly above the GMWL. The co-isotopic

plot reveals a large range of 7 ‰ in $\delta^{18}O$ from –30.4 ‰ to –23.4 ‰ and of 53 ‰ in $\delta D$ from –227 ‰ to –174 ‰. The respective slopes vary between 7.7 and 8.8 (Fig. 9 b).

## 5 Discussion

### 5.1 Cryolithological properties of the Sobo-Sise Yedoma IC and its Holocene cover

### 5.1.1 Permafrost aggradation rates and deposition history

Excluding the chronological gap between about 36 to 29 cal ka BP within unit A, we assume continuous permafrost aggradation of the MIS 3 Yedoma IC on Sobo Sise from at least about 52 to 36 cal ka BP, which is represented by a 16-m-thick permafrost sequence. The resulting aggradation rate of the MIS 3 Yedoma IC amounts to about 1 m per thousand years. The continuous permafrost aggradation assumed from MIS 2 Yedoma IC between about 28 and 20 cal ka BP excluding the hiatus within unit B (from 20 to 17 cal ka BP) formed a 5-m-thick sequence at a rate of about 0.7 m per thousand years. If

compared to the Bykovsky Yedoma IC record (site Mamontovy Khayata; Schirrmeister et al., 2002a), higher aggradation rates are obvious; about 1.5 m per thousand years for MIS 3 (12-m thick sequence between 46 and 38 cal BP) and about 0.85 m per thousand years for MIS 2 (6-m thick sequence between 28 and 21 cal ka BP). Thus, less permafrost aggradation during MIS 2 than during MIS 3 is also seen on Bykovsky Peninsula. However, it should be noted that the syngenetic growth of ice-oversaturated permafrost such as Yedoma IC is not only controlled by clastic and organic sedimentation, but further

triggered by formation of pore and segregation ice that contributes to Yedoma IC on Sobo-Sise 49 ± 10 wt% in MIS 3 and 43 ± 10 wt% in MIS 2 deposits. The volumetric ice content based on the absolute ice content (assuming ice saturation if the ice content is >20 wt%) according to Strauss et al. (2012) amounts to 66 ± 9 vol% and 65 ± 8 vol% for MIS 3 and MIS 2, respectively (Fuchs et al., under review). At this rather equal volumetric share of intrasedimental ice during MIS 3 and MIS 2, mainly organic accumulation seems to have controlled the difference in permafrost aggradation rates. It should further be

noted that growing ice wedges deform the frozen deposits in between by material transport from the polygon center toward the rim and upward push (Mackay, 1981). Thus, the vertical thickness of the sediment layers, determined now, might exceed the initial thickness due to the formation of intrasedimental (excess) ice, but also due to lateral material transport by the growing ice-wedges. The MIS 1 cover deposits accumulated the uppermost 1.4 m since about 6.4 cal ka BP. Due to freeze-thaw cycles in the active layer and thaw subsidence on the modern Sobo-Sise Yedoma surface of several centimeters per

year (Chen et al., 2018), the aggradation rate for unit C has not been calculated.

The MIS 3 Yedoma IC of unit A is characterised mainly by coarse silt and partly by fine sand. This is also seen in the prevalent rEM2 and rEM3 of unit A and differs from the bi-modal grain-size characteristics of the MIS 2 Yedoma IC of unit B with pronounced peaks in the coarse silt and medium sand fractions represented predominantly by rEM4 and rEM2 (Fig.

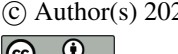



8; Fig. 10; Fig. S1). Such changes in grain-size distributions of MIS 3 and MIS 2 Yedoma IC may point to different material
sources and/or transport processes. A study by Schirrmeister et al. (2020) of Yedoma IC deposition history, sources and
material transport mechanisms includes the neighbouring study sites on Bykovsky Peninsula and Kurungnakh-Sise Island
(Fig. 1), but lacks a differentiation into MIS 3 and MIS 2 Yedoma IC as undertaken on data from Sobo-Sise. Therefore, a
direct comparison per formation period is of less use to disentangle changes in sedimentation over time, although some
general information can be deduced. The medium-sand rEM4 grain-size class of the Sobo-Sise data relates to high-energy
transport including saltation in meltwater runoff or fluvial water (rEM2 in Schirrmeister et al., 2020). The fine-sand rEM3
represents overbank deposits or settled suspensions in temporarily flooded sections during floodplain deposition (rEM4 in
Schirrmeister et al., 2020), while the coarse-silt rEM2 (rEM5 in Schirrmeister et al., 2020) might relate to floodplain
deposition as well, but could also originate from aeolian sources (Vandenberghe, 2013) or frost weathering processes
(Schwamborn et al., 2012). The fine-silt rEM1 (rEM8 in Schirrmeister et al., 2020) reflects low-energy settling of suspended
material from aeolian or pedogenic sources under still-water conditions, which is characteristic in low-center polygon ponds.
Thus, the Yedoma IC of Sobo-Sise formation during MIS 3 with prevailing fine sand (rEM3) and coarse silt (rEM2) derived
mainly from floodplain-related or meltwater runoff alluvial deposition processes, but possibly also includes aeolian and
frost-weathering components. The same coarse silt rEM2 dominates the grain-size distributions from MIS 2 deposits with a
pronounced peak (Fig. 8; Fig. 10; Fig. S1). We therefore assume a depositional regime similar to that in MIS 3 for this time.
MIS 2 deposition, however, shows a second pronounced peak in medium sand (rEM4) pointing to occasional high-energy
material transport in alluvial fan environments with strong meltwater runoff and/or fluvial transport. Sparse vegetation cover
as deduced from LGM climate conditions in the area (Andreev et al., 2011) might have promoted the potential for high
transport energy in a barren landscape. The differing grain-size compositions at the three location on Bykovsky, Sobo-Sise
and Kurungnakh-Sise reflect local diversity in accumulation processes for example with higher fluvial input on Kurungnakh-
Sise Island, but generally support the multi-process and multi-source regional Yedoma IC formation (Schirrmeister et al.,
2020).

### 5.1.2 Organic matter stocks and decomposition

The OM characteristics of the Sobo-Sise Yedoma IC differentiate into twofold higher organic carbon (TOC mean of 4.5±2.6
wt%) and 50% higher nitrogen (TN mean of 0.3±0.1 wt%) contents in unit A (MIS 3) if compared to those of unit B (MIS 2)
with mean TOC of 2.1±1.3 wt% and mean TN of 0.2±0.1 wt%. The resulting C/N ratios are slightly higher in unit A with
12.9 than in unit B with 10.5 (Table 1). A more productive tundra-steppe environment during MIS 3 (unit A) with higher
OM accumulation at comparable decomposition rates if compared to MIS 2 (unit B) is deduced.

Furthermore, unit A has significant higher carbon and nitrogen densities with a mean of 29±15 kg carbon m$^{-3}$ and 2.2±1.0 kg
nitrogen m$^{-3}$ compared to 14±8 kg carbon m$^{-3}$ and 1.4±0.4 kg nitrogen m$^{-3}$ in unit B. Consequently, the OM input into the
Lena River by fast erosion of the Yedoma cliff of Sobo-Sise (22.3 m yr$^{-1}$; Fuchs et al., under review) is mainly controlled by



unit A that stores twice the amount of carbon compared to unit B and which is exposed over about two thirds of the cliff height (Fig. 4).

The Holocene cover of unit C shows highest TOC (mean 11.3±9.9 wt%), TN (mean 0.6±0.3 wt%) and C/N (mean 18.5±8.0) of the record although of large variability (Table 1) mainly due to the low number of samples (n=4) in unit C. However, for
active layer samples in the Holocene cover layer of Sobo-Sise, Fuchs et al. (2018) detected mean values with high variability, too, with 6.7±7.4 wt%, 0.4±0.1 wt% and 15.8±12.3 for TOC, TN and C/N, respectively, indicating a general heterogeneity in OM accumulation in the uppermost soil layer. If compared to the neighbouring Yedoma IC sites on Bykovsky Peninsula (Schirrmeister et al., 2002a) and Kurungnakh-Sise Island (Schirrmeister et al., 2003; Wetterich et al., 2008a), the same pattern in OM properties over time from MIS 3 to MIS 1 supports regionally similar variations in
palaeoenvironmental conditions.

In fossil permafrost, the stable carbon and nitrogen isotope composition of organic matter is strongly controlled by the original botanical composition and further altered by decomposition (Weiss et al., 2016). The latter leads preferentially to loss of isotopically lighter $^{12}C$ and $^{14}N$ and thus enriches relatively the fraction of the heavier isotopes $^{13}C$ and $^{15}N$ by leaching and mineralisation processes (Tahmasebi et al., 2018). This fractionation towards less depleted isotopic carbon and
nitrogen compositions over time occurs before the OM enters the perennially frozen state. Thus, the permafrost aggradation rate during distinct periods further influences the rate of OM decomposition. However, the differences seen per units in the Sobo-Sise Yedoma record are minor for $\delta^{15}N$ with mean values of around 2 ‰ for all three units (Table 1, Fig. 11). The $\delta^{13}C$ unit mean values vary over about 2 ‰ (between about –28 ‰ and –26 ‰), and are most depleted for unit C (MIS 1). Due to these only little variations and the range overlap, no significant differences in OM decomposition can be interpreted from the
stable carbon and nitrogen isotope composition for the three units.

### 5.1.3 Intrasedimental ice characteristics

Highest DOC concentrations up to 754 mg L$^{-1}$ in MIS 3 together with highest average C/N ratios indicate OM preservation (Fig. 5; Fig. 6). Rapid sediment and OM accumulation rates, as indicated by radiocarbon-based age-depth relationship, lead to effective syngenetic permafrost formation so that particulate and dissolved OM are rapidly incorporated into permanently
frozen deposits. Hence, OM degradation is minimised and labile or soluble DOM fractions have not been drained or flushed out from rapidly aggrading permafrost.

The stable water iotope ($\delta^{18}O$ and $\delta D$) and major ion compositions as well as DOC concentrations of Sobo-Sise intrasedimental ice reflect the general cryostratigraphy and have palaeoclimate implications. Preservation of pore water during formation of segregated ice occurs via a wide range of processes. Nevertheless, several studies (e.g. Mackay, 1983;
Murton and French, 1994; Kotler and Burn, 2000; Schwamborn et al., 2006; Fritz et al., 2012) have shown that $\delta^{18}O$ and $\delta D$ isotopes in intrasedimental ice can still reflect environmental and climatic changes when considered with caution and/or focused on pore ice (Porter et al., 2019; Porter and Opel, 2020). Higher $\delta^{18}O$ and $\delta D$, but lower $d$ values are found in MIS 3





compared to MIS 2 (Fig. 6; Fig. 9). Relatively warm summers during the MIS 3 interstadial might explain the lower *d* values in associated intrasedimental ice due to a higher water loss by evaporation (i.e., kinetic fractionation). This would lead to a

water reservoir in polygon ponds and soil moisture that becomes successively depleted in $^{16}$O and $^{1}$H compared to the original precipitation. Increased temperature and precipitation amplitudes during MIS 3 (Andreev et al., 2011; Pitulko et al., 2017) may have led to frequent drying and re-wetting in polygon tundra and thus to enhanced kinetic fractionation. Another process of kinetic fractionation producing the same pattern are multiple freeze-thaw cycles of soil moisture in the active layer (Throckmorton et al., 2016).

Elevated ion (Mg, Ca, Na, Cl) concentrations with EC up to 5800 μScm$^{-1}$ in the MIS 3 record (unit A; Fig. 6) are likely caused by frequent drying and re-wetting in polygonal tundra in times of higher summer temperature and precipitation amplitudes during the interstadial compared to MIS 2 stadial (unit B). Meyer et al. (2002a) found similarly elevated EC values of 5500 μScm$^{-1}$ in MIS 3 deposits on Bykovsky Peninsula. Modern surface waters in Central Yakutia at high continentality show EC values of up to 5710 μScm$^{-1}$ (Wetterich et al., 2008b) and even up to 7744 μScm$^{-1}$ (Pestryakova et

al., 2018). Ion-rich pore waters have also been found in MIS 3 deposits at Buor Khaya Peninsula (Schirrmeister et al., 2017), but with different composition and including a distinct saline horizon. In contrast, ion composition in the Sobo-Sise Yedoma IC remained stable throughout MIS 2 and MIS 3 and is dominated by Mg, Cl and Ca in both units. Therefore, we assume that water and sediment sources did not change over time, but reflect higher evaporation during warmer summers in MIS 3 if compared to MIS 2.

**5.2 Palaeoclimatic implications from regional wedge-ice records**

Sobo-Sise ice wedge stable isotopes show a complex pattern that at least in parts can be related to the fact that Holocene ice wedges formed epigenetically within older late Pleistocene deposits and penetrated pre-existing ice wedges. This may be related to subsidence and thermo-erosional processes that thaw permafrost, lower the surface and complicate the stratigraphic attribution of the wedge ice. The stable isotope composition of ice wedge profiles sampled in the central

(SOB18-02) and western parts (SOB18-08) of the Sobo-Sise cliff allows differentiating late Pleistocene and Holocene wedge ice.

Generally, late Pleistocene wedge ice is characterised by well-depleted δ$^{18}$O and δD values (mean values between –30 ‰ and –29 ‰, and –232 ‰ and –225 ‰, respectively; Fig. 9 c) and low *d* values (means between 5‰ and 7‰; Table 2). In contrast, a striking feature of Holocene ice wedges are their significantly elevated mean *d* values between 11‰ and 15‰

(Table 2) accompanied by surprisingly low δ$^{18}$O and δD values (mean values between –28 ‰ and –25 ‰, and –207 ‰ and – 190 ‰, respectively; Fig. 9 b). In some instances, Holocene δ$^{18}$O and δD values reach and even exceed the values of the late Pleistocene ice wedges (Table 2). This is true for both the oldest Holocene ice wedge stage, i.e. the toes of ice wedge SOB14-IW5 at the Ice Complex Alas slope and the late Holocene to modern ice wedges on the top of the Ice Complex (e.g. SOB14-IW4). Hence, the isotopic difference between late Pleistocene and Holocene ice wedges is more pronounced in *d*

than in δ values. It is obvious that the horizontal profiles of late Pleistocene ice wedges are less spiky than those of Holocene



ice wedges (Fig. 7). This might indicate a time-dependent smoothing of the isotope profiles due to isotopic diffusion within ice wedges but is beyond the scope of this study.

All co-isotopic regression slopes are highly correlated ($R^2 > 0.97$) and vary between 7.16 and 9.43 (Table 2). While the ice wedges of units C (MIS 3) and A (MIS 1) show relatively coherent patterns, the unit B ice wedge SOB18-02-I sticks out

with a value of 9.43 (Fig. 9 b and c), likely related to a comparably low internal isotope variability. Hence, we assume that the isotopic composition of all ice wedges carry paleoclimate information for the winter season and is not significantly altered by secondary fractionation processes.

The Sobo-Sise ice-wedge stable isotopes of units A and B fit mostly well into the regional pattern of the Central Laptev Sea coast and the Lena Delta (Fig. 1) as well as in the large-scale pattern as presented by Opel et al. (2019) and reflect cold and

stable winter climate conditions during the last glacial. Unit A (MIS 3) Sobo-Sise ice wedges (mean $\delta^{18}O$: –29.7‰, mean $\delta D$: –231.8‰) are slightly less depleted compared to those of Bykovsky Peninsula (mean $\delta^{18}O$: –30.8‰, mean $\delta D$: –242.8‰) to the east and Kurungnakh-Sise Island (mean $\delta^{18}O$: –31.6‰, mean $\delta D$: –247.6‰) to the west (Opel et al., 2019). In contrast, ice-wedge $d$ values are slightly higher at Sobo-Sise (mean $d$: 5.7‰) compared to Bykovsky (mean $d$: 3.7‰) and Kurungnakh-Sise (mean $d$: 5.3‰). For ice wedges of the MIS 2, the pattern is similar with slightly less depleted $\delta$ values for

Sobo-Sise (mean $\delta^{18}O$: –28.8‰, mean $\delta D$: –224.6‰) and slightly higher $d$ values (mean $d$: 7.4‰) compared to Bykovsky (mean $\delta^{18}O$: –30.6‰, mean $\delta D$: –239.5‰, mean $d$: 5.1‰, Meyer et al., 2002a) while no data are available for Kurungnakh-Sise. As the sampled ice wedges likely cover different parts of the MIS 3 and MIS 2 Yedoma IC and therefore different time slices, the slight differences should not be spatially interpreted in terms of winter temperature differences.

The stable isotope compositions of the studied Sobo-Sise ice wedges do not show any significant differences between ice

wedges of units A and B, corresponding to MIS 3 and 2, respectively. This might indicate that the globally cold LGM is not reflected in the Sobo-Sise ice wedge-based winter climate record and would be in accordance with both regional scale, when compared to Bykovsky Peninsula (Meyer et al., 2002a) or to other study sites in the Laptev Sea region (Wetterich et al., 2011), and also Arctic-wide scale (Porter and Opel, 2020). It is not sufficiently resolved yet, whether this is due to a less cold LGM climate in the region or whether the LGM cold period is not captured by the studied ice-wedge profiles that do not

preserve a continuous record. In this context, we observe (1) a depositional gap temporally coinciding to peak LGM conditions for the three sites at regional scale and (2) extremely depleted LGM ice-wedge isotopes have been only found at Bol'shoy Lyakhovsky Island further east (Fig. 1; Wetterich, et al., 2011).

In accordance with ice-wedge records at Bykovsky and Kurungnakh-Sise, the Holocene Sobo-Sise ice wedges of unit C show distinctly warmer winters and significantly changed moisture generation pattern compared to the late Pleistocene

records. Overall Holocene mean ice wedge $\delta$ values on Sobo-Sise are enriched by about 1.8 ‰ to 2.7 ‰ for $\delta^{18}O$ and 23 ‰ to 30 ‰ for $\delta D$, over MIS 2 and MIS 3 ice wedges, respectively. Mean Holocene ice-wedge $d$ value (14.2 ‰) is about 7 ‰ and 8 ‰ higher compared to MIS 2 and MIS 3, respectively, indicating substantial changes in the moisture generation and transport patterns (e.g. Meyer et al., 2002a). Similar changes have been observed on Bykovsky Peninsula (Meyer et al., 2002a), while Holocene ice wedges at Kurungnakh-Sise show more enriched mean $\delta$ values and lower mean $d$ values





(Schirrmeister et al., 2003; Wetterich et al., 2008a). It has to be noted that the Holocene ice wedge stable isotope compositions for both Sobo-Sise and Bykovsky exhibit significantly more depleted δ values and significantly higher *d* values compared to other ice-wedge study sites along the Siberian Arctic coastal lowlands (Opel et al., 2019). Holocene minimum δ values even fit well into the typical MIS 3 and MIS 2 isotopic range. It is, however, unlikely, that this particular region, i.e. the eastern Lena River Delta and the western Tiksi Bay, is characterised by a significantly colder winter climate. Hence,

other potential explanations have to be considered, such as regional specifics of the water cycle. The significantly depleted δ values and increased *d* values of Holocene ice wedges show some similarities to early winter precipitation (October to December) that is, in particular characterised by distinctly increased *d* values (e.g. Kurita, 2011; Bonne et al., 2020). Hence, the Holocene ice wedge stable isotope composition might be explained by an over-representation of early winter snow to the melt water feeding ice-wedge cracks. This could be related to specific moisture generation and transport patterns influencing

the precipitation in this particular region. A second option could be the contribution of moisture from local sources such as evaporation of isotopically depleted and high deuterium excess Lena River water (Juhls et al., 2020) in the period of ice build-up, resulting in a substantial snow cover development in the early winter season. An only little mixed ocean and substantial open water areas with mainly freshwater signature could explain why this low δ and high *d* values pattern for Holocene ice wedges could so far only be observed in the eastern Lena Delta region.

**5.3 Chronostratigraphy of the Yedoma IC in the Central Laptev Sea coastal region**

The geochronological record of the Sobo-Sise Yedoma IC spans the last about 52 cal ka BP based on the stacked age-height modelling. Older parts of the Yedoma IC are likely to be found up to several meters below the modern river level (Fuchs et al., under review) as it has also been reported from other sites in the eastern Lena Delta (Pavlova and Dorozhkina, 2000) and from Bykovsky Peninsula (Schirrmeister et al., 2002b). The lowermost sample of the Sobo-Sise record had a finite age of 47

021 ± 646 BP (SOB18-06-35) but is beyond the limit of calibration. However, it supports the modelled age range of the record down to 51.8 cal ka BP. The entire record exhibits three substantial chronological gaps, which are from about 36.7 to 28.4 cal ka BP, from about 20.4 to 16.8 cal ka BP and from about 15.5 to 6.4 cal ka BP (Fig. 12).

Taking into account that the exposure conditions and the applied sampling and dating resolution largely define the quality of the resulting geochronological record, we compare our Yedoma IC dataset from Sobo-Sise Island (32 [14]C dates over 24 m

profile length) to similar ones with a resolution of largely better than 1 m in vertical dimension, i.e. those from site Mamontovy Khayata at Bykovsky Peninsula (51 [14]C dates over 37 m profile length; Schirrmeister et al., 2002a; Grosse et al., 2007) and Kurungnakh-Sise Island in the central Lena Delta (19 [14]C dates over 19 m profile length; Schirrmeister et al., 2003; Wetterich et al., 2008a). However, the sampling approaches differed. On Bykovsky and Kurungnakh-Sise the exposures were sampled during different years at highly dynamic thaw slumps over a rather large lateral extent, i.e. up to

several hundreds of meters. Exposed baidzherakhs (thaw mounts of former polygon centers) at different height levels were sampled. In contrast, the permafrost sampling at the vertical Yedoma cliff on Sobo-Sise was performed in three closeby (i.e. within about 120 meters) overlapping profiles likely resulting in complete coverage of the exposed permafrost sequence.



If compared to nearby studied Yedoma profiles to the east on Bykovsky Peninsula in the Central Laptev Sea and to the west on Kurungnakh-Sise Island in the central Lena Delta a similar pattern is striking. In detail, the Bykovsky record spans from
about 60 ka BP and shows the smallest gaps of all considered records from 38 to 32.5 cal ka BP, from 21 to 18 cal ka BP and from 12.5 to 9 cal ka BP (Fig. 12). The Kurungnakh-Sise record shows two large age gaps from 37 to 21 cal ka BP and from 20 to 9 cal ka BP (Fig. 12) found in two independent sampling campaigns (Schirrmeister et al., 2003; Wetterich et al., 2008a).

The hiatus overlap recognised at all three Yedoma sites studied in the region, i.e Bykovsky, Sobo-Sise and Kurungnakh-Sise
(Fig. 1), results in three gaps of likely overarching relevance that are found during MIS 3 from 36 to 32.5 cal ka BP, during MIS 2 from 20.5 to 18 cal ka BP and during MIS 2-1 transition from 12.5 to 9 cal ka BP (Fig. 12).

To explain the observed gaps in the chronological records, two mechanisms need to be discussed that are (1) no or extremely low deposition during a certain period of time, and/or (2) thaw and erosion of a certain sequence after deposition. Both mechanisms might be related to a variety of processes spanning from global to regional climate variations over time to local
geomorphologic disturbance processes that are not necessarily or solely climate-triggered. To disentangle the general hiatus of three time periods at three Yedoma IC sites in the Lena-Laptev region, the following discussion lines can be drawn.

### 5.3.1 Interstadial climate variability and consecutive local disturbance vs. fluvial erosion during MIS 3

The proposed regional overlap hiatus in MIS 3 Yedoma IC deposits spans 3 500 years (36 - 32.5 cal ka BP, Fig. 12). The interstadial climatic variability during MIS 3 deduced from permafrost sequences of NE Siberia was subject to previous
studies which assume a MIS 3 climatic optimum expressed by warm summer conditions; mainly based on botanic proxy data (e.g. Anderson and Lozhkin, 2001; Andreev et al., 2011; Murton et al., 2015, 2017; Pitulko et al., 2017). The proxy record, however, varies in both duration and timing at different Yedoma IC study sites from the Western Laptev coast to the Kolyma lowland (see Fig. 11 in Wetterich et al., 2014). If warmer summer climate conditions during MIS 3 interstadial led to partial Yedoma IC thaw and to the observed overall depositional gap at 36-32.5 cal ka BP by deepening of the active layer and
related surface subsidence resulting in degradation rates exceeding aggradation rates, the timing of the MIS 3 climatic optimum is of specific interest. The close-by Yedoma IC site on Bykovsky Peninsula allows for proxy-based reconstruction of MIS 3 interstadial environmental conditions. Here, warm conditions with mean summer temperatures >12 °C and the occurrence of standing water are deduced for around 40-39 cal ka BP from fossil findings of e.g. *Callitriche hermaphroditica* that is a temperate aquatic plant (Kienast et al., 2005) supported by findings of diverse ostracod faunae that
inhabited low center polygon ponds during the same time period (Wetterich et al., 2005). However, all evidence from Bykovsky records for MIS 3 climate optimum predate the observed hiatus by about 7 000 to 6 000 years which makes it unlikely that the observed MIS 3 gap was driven by regional climate. The MIS 3 hiatus in the Kurungnakh-Sise Yedoma IC spans even more from about 37 to 21 cal ka BP - missing substantial parts of the MIS 3 (Schirrmeister et al., 2003; Wetterich et al., 2008a), but again only after the supposed MIS 3 climatic optimum that is likewise in the Bykovsky Yedoma





paleontological record reflected by warm summer conditions and the presence of low-centered polygon tundra providing a broad landscape mosaic of ecological niches for e.g. insects and plants indicating dry and warm conditions in drained positions and for aquatic organisms inhabiting polygon ponds (Khazin et al., 2019; Wetterich et al., 2008a). The observed MIS 3 hiatus in the Bykovsky, Sobo-Sise and Kurungnakh Yedoma IC records is not during the MIS 3 climatic optimum, but might be related to general MIS 3 climate instability as expressed in Dansgaard/Oeschger (D/O) events recorded in the

Greenland ice cores (e.g., Dansgaard et al., 1993; NGRIP members, 2004). But it is unknown whether D/O events have impacted the East Siberian Arctic and the lack of deposits or coarse chronology in the Yedoma IC records prevents to draw conclusion on a linkage to these palaeoclimatic events.

Another possible explanation for the MIS 3 hiatus is proposed by Margold et al. (2018) who found evidence for repeated cataclysmic outburst floods from the glacial Lake Vitim in Southern Siberia into the Vitim River valley and further into the

Lena River valley towards the Arctic Ocean. The flooding events were dated by multiple techniques including optically-stimulated luminescence dating and cosmogenic nuclide dating (Be-10 bedrock exposures and Be-10 depth profiles). The reconstructed flood chronology spans over the last 60 ka of which the timing of megaflood II at around 34 ka fits into the chronologic gap observed in the Yedoma IC chronology of Bykovsky, Sobo-Sise and Kurungnakh around 36-32.5 cal ka BP (Fig. 12). Thus, fluvial impact by the proposed megaflood event might have affected the continuity of the Yedoma IC

chronologies by eroding considerable parts of the sequences. But except for the chronology gaps no direct erosional features such as fluvial sand or pebble layers have been observed in the outcrops. Thus, the flood may have only changed the hydrological regime (e.g. by developing new discharge paths similar to the channels in the today's Lena Delta) for a certain time period and by doing so prevented the deposition of fluvially transported material in the areas of the studied Yedoma IC outcrops. If so, this could have stopped or minimised Yedoma IC accumulation at the study sites.

**5.3.2 LGM climate vs. fluvial erosion during MIS 2**

The second distinct overlapping hiatus in the Yedoma IC chronologies of Bykovsky Peninsula, Sobo-Sise and Kurungnakh-Sise islands occurred during MIS 2 at 20.5-18 cal ka BP (Fig. 12), and falls partly in the last glacial maximum (LGM) period around 26.5-19 cal ka BP (Clark et al., 2009). The LGM environments of the Laptev Sea coastal region are characterised by cold and dry summer conditions and represented in pollen records by grass-dominated communities with Caryophyllaceae,

Asteraceae, Cichoriaceae, *Selaginella rupestris* (Andreev et al., 2011). Further paleontological evidence for cold and dry summers is provided by plant macrofossils and insect fossil records from the Bykovsky Yedoma IC (Kienast et al., 2005; Sher et al., 2005) while the LGM is almost not captured in the Kurungnakh-Sise Yedoma IC record (Schirrmeister et al., 2003; Wetterich et al., 2008a). The less productive summer conditions most likely hampered OM accumulation while reduced ice-wedge growth might be related to less winter precipitation and stronger snow drift by stronger wind activity

affecting snow drift and sublimation; both leading to reduced Yedoma IC formation during MIS 2 if compared to MIS 3 as also seen in the lower permafrost aggradation rate (see section 5.1.2). However, no permafrost aggradation during MIS 2 at all seems unlikely in the larger study region since it has a good depositional representation in several Yedoma IC sites



(Duvanny Yar - Murton et al., 2015; Yana lowland, Pitulko et al., 2004, 2017; Bol'shoy Lyakhovsky - Wetterich et al., 2011; Mamontov Klyk - Schirrmeister et al., 2008; Fig. 1). On Bol'shoy Lyakhovsky Island (north-east of the Central Laptev Sea

region) a shift from accumulation on top of the MIS 3 Yedoma IC to valley positions was found and explained by a lowered erosion base due to LGM sea level lowstand and according changes in the hydrological system of areas with higher relief inclination (Wetterich et al., 2011). As seen in Fig. 12, the MIS 2 chronologic time gap is larger in the central Lena Delta (about 11 ka between about 20 and 9 cal ka BP on Kurungnakh-Sise) if compared to the eastern Lena Delta (about 3 ka between about 20 and 17 cal ka BP on Sobo-Sise) and to Bykovsky Peninsula (about 3 ka between about 21 and 18 cal ka

BP at Mamontovy Khayata, site no. 1 in Fig. 1). Additionally, Grosse et al. (2007) reported an observation at the northern end of Bykovsky Peninsula where 22-m thick MIS 3-2 Yedoma IC (dated from about 53 ka BP to 23 cal ka BP) is discordantly covered by 3-m thick sand with organic interlayers of probably shallow fluvial origin dated to about 16 cal ka BP (site B-S in Grosse et al., 2007; site no. 2 in Fig. 1). For the large gap in the Kurungnakh-Sise MIS 2 Yedoma IC record it might also be possible that the MIS 2 gap likely induced by fluvial erosion of megaflood III (Margold et al., 2018) further

combines with the deglacial (MIS 2-1) gap and any possible deposition in between got eroded by the latter.

### 5.3.3 Deglacial thermokarst during MIS 2-1

The global lateglacial to early Holocene warming manifested the transition from glacial to interglacial conditions. The effect of warming on permafrost conditions is largely captured by an increase in ground temperature, a deepening of the seasonally thawed active layer, surface subsidence, and activation of thermokarst and thermo-erosional processes (e.g., Wetterich et al.,

2009). The resulting ground ice melt and permafrost thaw led to re-organisation of the post-Beringian periglacial landscapes and accumulation areas remaining after the opening of the Bering Strait around 11 cal ka BP (Jakobsson et al., 2017) and the subsequent Holocene sea-level rise and shelf inundation (Bauch et al., 2001; Klemann et al., 2015). The large-scale warming pulse terminated the accumulation of the Yedoma IC during the lateglacial period as it is seen in the age gaps of the Yedoma IC records considered here (Fig. 12) leading to an overlap hiatus at 12.5-9 cal ka BP. Similar lateglacial-Holocene hiatus are

found for many other chronostratigraphic records of Yedoma ICs (Fig. 1) such as in the Kolyma lowland at the Duvanny Yar site (Murton et al., 2015), on the New Siberian Islands (Schirrmeister et al., 2011a; Wetterich et al., 2009, 2014), on Buor Khaya Peninsula (Schirrmeister et al., 2017) and on Mamontov Klyk (Schirrmeister et al., 2008). During the lateglacial to early Holocene warming intense thermokarst within degrading Yedoma IC created new accumulation areas, i.e. thermokarst basins and thermo-erosional valleys, which dominate the modern surface morphology in Arctic lowlands by more than 50 %

of the modern surface on Bykovsky Peninsula (Grosse et al., 2005; Fuchs et al., 2018), on Sobo-Sise (Fuchs et al., 2018) and on Kurungnakh-Sise islands (Morgenstern et al., 2011).

Dated records of thermokarst deposition commonly fit into the hiatus that represents the end of Yedoma IC formation. Lateglacial to early Holocene thermokarst deposits on Bykovsky Peninsula are dated from about 10 to 1 cal ka BP (Schirrmeister et al., 2002a), on Kurungnakh-Sise Island from about 15 cal ka BP to modern (Morgenstern et al., 2013) and

on Sobo-Sise Island from about 7.4 cal ka BP to modern (Fuchs et al., 2018). Thermo-erosional valleys as erosional features



of Yedoma IC degradation were dated on Bykovsky from 5 to 1 cal ka BP (Schirrmeister et al., 2002a). However, Holocene cover deposits on top of Yedoma IC are common and also observed on Sobo-Sise where they were dated from 9.8 to 1.3 cal ka BP (Fuchs et al., 2018) and from 6.4 to 2.4 cal ka BP (unit C in this study). In summary, overall climate warming at the transition from glacial to interglacial conditions promoted extensive Yedoma IC thaw and created new accumulation areas in
thermokarst basins and thermo-erosional valleys. Both, the IC degradation and the change in deposition processes caused the hiatus on top of the Yedoma IC of the Laptev Sea coastal region.

**6 Conclusions**

Late Pleistocene permafrost of the Yedoma Ice Complex type is widespread in the East Siberian Arctic, but sediment sequences are often discontinuous due to (1) the vulnerability of ice-rich permafrost to thaw under warming conditions, (2)
surface erosion in times of arid and windswept conditions, and (3) periglacial processes such as cryoturbation and internal reorganisation of polygonal landscapes. We identified three different cryostratigraphic units at the Sobo-Sise Yedoma IC. Unit A (52–28 cal ka BP) represents the depositional environment during interstadial MIS 3 and is characterised by coarse silt and fine sand, while unit B (28–15 cal ka BP) representing the stadial MIS 2 conditions is dominated by coarse silt and middle sand and lower organic matter (carbon, nitrogen) contents compared to unit A. In addition, unit A has higher
permafrost aggradation rates (1 m ka$^{-1}$) compared to unit B (0.7 m ka$^{-1}$). The sedimentary properties of both units support the hypothesis of multi-process and multi-source regional Yedoma IC formation, here mainly triggered by ice-wedge polygon formation under floodplain conditions with varying input shares of cryogenic, fluvial, pedogenic and aeolian origin. The Yedoma IC is discordantly overlain by organic-rich Holocene deposits of unit C (7–0 cal ka BP).

The wedge-ice records cover all three cryostratigraphic units as shown by radiocarbon-dated organic matter from inside the
ice. The stable isotope composition of the Sobo-Sise wedge ice in comparison to other regional records shows similar patterns with, e.g. no indication of LGM cold period in MIS 2 wedge ice. Thus, MIS 3 and MIS 2 ice wedges have rather similar isotopic compositions as also observed on Bykovsky Peninsula. A further regional peculiarity is low $\delta^{18}$O and high deuterium excess in Holocene records that might represent a contribution of regional isotopically depleted water source, e.g. Lena water.

The chronostratigraphy of the Sobo-Sise cliff revealed three hiatus at about 36–29 cal ka BP, 20–17 cal ka BP and 15–7 cal ka BP, which are in accordance with hiatus from two other Yedoma IC deposits in close vicinity on Bykovsky Peninsula and Kurungnakh-Sise Island. Similar patterns but different duration found on these locations indicate a regional signal of disturbance leading to either low accumulation or erosion of deposited material. We hypothesize that the first two regional hiatus overlaps (36–32.5 cal ka BP and 20.5–18 cal ka BP) are related to megafloods proposed by Margold et al. (2018),
although more evidence is needed to confirm this hypothesis. The last overlap hiatus in the regional chronostratigraphy (12.5–9 cal ka BP) is caused by climate-driven permafrost thaw and consecutive change of accumulation areas during the lateglacial to Holocene transition as it was observed in many other Yedoma IC deposits in north-eastern Siberia. Thus, the



Sobo-Sise Yedoma record represents a rather typical example of late Pleistocene Ice Complex formation under West Beringian conditions superimposed in its preservation by thaw events that were fluvially-triggered during MIS 3-2 and
climate-triggered during MIS 2-1.

**Data availability**

Original data will be available at PANGAEA after final acceptance of the paper: https://doi.org/10.1594/PANGAEA.919470, 2020 (Wetterich et al., 2020).

**Author contributions**

SW conceptualised the research. SW, MiFr, TO and LS designed the fieldwork, which was performed with help of AK and AA. AK performed the climbing and sediment sampling of the Yedoma cliff, while SW, MiFr, TO and LS performed the wedge-ice sampling. Laboratory work and data analyses were carried out by SW, MiFr, TO, HaMe, LS, GM, JW, JR, MaFu and HeMa. SW wrote the manuscript with input from all co-authors.

**Competing interests**

The authors declare that they have no conflict of interest.

**Acknowledgements**

The fieldwork of this study received great logistic support in summer 2018 from the Hydrobase Tiksi (Dmitry Mel'nichenko) and AWI logistics (Expedition LENA 2018, Volkmar Assmann and Waldemar Schneider). The laboratory analyses were expertly conducted by Antje Eulenburg, Mikaela Weiner, Lutz Schönicke and Dyke Scheidemann (AWI
Potsdam) as well as by Elizabeth Bonk and Torben Gentz (MICADAS, AWI Bremerhaven). Ingmar Nitze (AWI Potsdam) helped retrieving the cliff edge line for Figure 2 and Janet Rethemeyer (CologneAMS) provided three radiocarbon dates.

**Financial support**

This research has been supported by Deutsche Forschungsgemeinschaft (DFG grant no. WE4390/7-1 to SW and grant no. OP217/4-1 to TO), by the EU Framework Programme for Research and Innovation - Horizon 2020 (grant agreement number
773421; project: NUNATARYUK to MiFr), by the NERC-BMBF project CACOON (grant no. 03F0806A, Changing Arctic Ocean (CAO) program to MaFu) and the Russian Foundation for Basic Research (RFBR grant no.18-05-60080 to AK).



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





**Tables**

Table 1: Data of the three cryostratigraphic units A, B and C of the Sobo-Sise Yedoma cliff summarising minimum (MIN), mean (MEAN), and maximum (MAX) values as well as standard deviations (SD) of sedimentological, organic-matter, stable-isotope and hydrochemical analyses.

| | Unit C (MIS 1 Holocene cover) | | | | | Unit B (MIS 2 Yedoma IC) | | | | | Unit A (MIS 3 Yedoma IC) | | | | |
|---|---|---|---|---|---|---|---|---|---|---|---|---|---|---|---|
| | N | MIN | MEAN | MAX | SD | N | MIN | MEAN | MAX | SD | N | MIN | MEAN | MAX | SD |
| Mean grain-size [µm] | 4 | 53 | 66 | 82 | 13 | 16 | 52 | 113 | 303 | 64 | 40 | 28 | 46 | 75 | 12 |
| | 4 | 2.2 | 2.4 | 2.5 | 0.1 | 16 | 2.0 | 2.4 | 2.8 | 0.2 | 40 | 1.9 | 2.1 | 2.3 | 0.1 |
| MS [HF] | 4 | 19 | 32 | 66 | 23 | 16 | 31 | 53 | 71 | 9 | 41 | 23 | 40 | 58 | 9 |
| TN [wt%] | 4 | 0.2 | 0.6 | 0.8 | 0.3 | 14 | 0.1 | 0.2 | 0.4 | 0.1 | 41 | 0.2 | 0.3 | 0.7 | 0.1 |
| TC [wt%] | 4 | 3.2 | 12.5 | 27.1 | 10.3 | 16 | 0.7 | 2.6 | 5.4 | 1.3 | 41 | 2.5 | 5.0 | 15.6 | 2.7 |
| TOC [wt%] | 4 | 3.0 | 11.3 | 25.5 | 9.9 | 15 | 0.5 | 2.1 | 5.1 | 1.3 | 41 | 1.7 | 4.5 | 15.1 | 2.6 |
| TOC/TN | 4 | 13.8 | 18.5 | 30.6 | 8.0 | 14 | 7.2 | 10.5 | 13.6 | 2.4 | 41 | 9.1 | 12.9 | 21.7 | 2.5 |
| $\delta^{13}C$ [‰] vs. PDB | 4 | −28.32 | −28.01 | -27.76 | 0.24 | 15 | −27.37 | −26.07 | −25.22 | 0.59 | 41 | −29.89 | −27.29 | −25.97 | 0.91 |
| $\delta^{15}N$ [‰] vs. AIR | 4 | 1.26 | 2.14 | 2.89 | 0.69 | 14 | 0.54 | 1.91 | 3.19 | 0.97 | 41 | 1.20 | 2.24 | 3.69 | 0.57 |
| Ice content [wt%] | 3 | 41 | 56 | 80 | 21 | 16 | 20 | 43 | 61 | 10 | 41 | 24 | 49 | 74 | 10 |
| $\delta^{18}O$ [‰] vs. SMOW | 3 | −20.98 | −20.70 | −20.36 | 0.32 | 15 | −28.69 | −26.21 | −21.58 | 2.17 | 35 | −28.24 | −23.88 | −20.27 | 2.00 |
| $\delta D$ [‰] vs. SMOW | 3 | −152.9 | −150.7 | -148.6 | 2.2 | 15 | −216.6 | −199.9 | −158.8 | 16.1 | 35 | −221.8 | −189.6 | −163.5 | 14.5 |
| deuterium excess [‰] vs. SMOW | 3 | 14.3 | 14.9 | 15.5 | 0.6 | 15 | 3.1 | 9.7 | 15.3 | 3.9 | 35 | -6.0 | 1.5 | 12.0 | 3.9 |
| DOC [mg L⁻¹] | 1 | n/a | 33.7 | n/a | n/a | 8 | 85.3 | 212.4 | 588.7 | 159.9 | 20 | 160.9 | 366.8 | 753.5 | 174.0 |
| EC [µS cm⁻¹] | 1 | n/a | 36 | n/a | n/a | 4 | 1129 | 1806 | 3180 | 950 | 12 | 726 | 2245 | 5790 | 1568 |
| Chloride [mg L⁻¹] | 1 | n/a | 1.5 | n/a | n/a | 4 | 61.8 | 340.8 | 873.1 | 370.0 | 12 | 8.3 | 494.9 | 1631.9 | 534.8 |
| Sulfate [mg L⁻¹] | 1 | n/a | 0.7 | n/a | n/a | 4 | 2.5 | 101.2 | 282.1 | 132.2 | 12 | 3.2 | 72.7 | 255.1 | 80.8 |
| Ca [mg L⁻¹] | 1 | n/a | 3.0 | n/a | n/a | 4 | 48.5 | 160.2 | 334.0 | 123.0 | 12 | 75.4 | 147.0 | 265.5 | 61.7 |
| Fe [mg L⁻¹] | 1 | n/a | 3.2 | n/a | n/a | 2 | 0.7 | 3.2 | 5.6 | 3.5 | 8 | 0.1 | 6.0 | 37.4 | 12.7 |
| K [mg L⁻¹] | 1 | n/a | 0.8 | n/a | n/a | 4 | 3.0 | 5.8 | 8.5 | 2.5 | 12 | 3.2 | 7.6 | 11.7 | 2.9 |
| Mg [mg L⁻¹] | 1 | n/a | 1.4 | n/a | n/a | 4 | 74.7 | 110.5 | 164.3 | 39.3 | 12 | 48.8 | 144.9 | 372.3 | 89.8 |
| Mn [mg L⁻¹] | 1 | n/a | 0.3 | n/a | n/a | 4 | 0.2 | 1.6 | 3.8 | 1.6 | 12 | 0.3 | 1.3 | 2.7 | 0.7 |
| Na [mg L⁻¹] | 1 | n/a | 1.7 | n/a | n/a | 4 | 38.9 | 60.9 | 92.8 | 22.9 | 12 | 24.4 | 105.9 | 485.2 | 133.4 |






**Table 2: Stable-isotope data ice-wedges of the Sobo-Sise Yedoma cliff summarising minimum (MIN), mean (MEAN), and maximum (MAX) values as well as standard deviations (SD). Radiocarbon dates of organic material from inside the wedge ice are further given in Table 3.**

| Units | Age range [cal BP] | ID | $\delta^{18}O$ [‰] vs. SMOW | | | | $\delta D$ [‰] vs. SMOW | | | | deuterium excess [‰] vs. SMOW | | | | N | Slope | Intercept | R² |
|---|---|---|---|---|---|---|---|---|---|---|---|---|---|---|---|---|---|---|
| | | | MIN | MEAN | MAX | SD | MIN | MEAN | MAX | SD | MIN | MEAN | MAX | SD | | | | |
| C | n/a | SOB14-05 | −28.66 | −27.28 | −25.85 | 0.74 | −215.2 | −204.4 | −193.6 | 5.8 | 12.3 | 13.8 | 15.0 | 0.8 | 33 | 7.73 | 6.55 | 0.98 |
| C | n/a | SOB18-08-II | −29.06 | −26.18 | −24.42 | 1.09 | −218.7 | −195.7 | −180.5 | 9.0 | 11.5 | 13.8 | 16.1 | 0.7 | 112 | 8.20 | 19.05 | 0.99 |
| C | modern–2 290 | SOB14-04 | −30.43 | −27.77 | −24.06 | 1.73 | −226.5 | −207.1 | −178.0 | 13.7 | 13.1 | 14.9 | 17.0 | 0.7 | 131 | 7.90 | 12.24 | 1 |
| C | n/a | SOB18-02-II | −27.13 | −25.18 | −23.38 | 1.27 | −207.0 | −190.3 | −173.9 | 11.1 | 9.3 | 11.2 | 13.2 | 1.1 | 15 | 8.75 | 30.04 | 1 |
| B | 23 470–25 350 | SOB18-02-I | −29.37 | −28.80 | −27.75 | 0.47 | −230.3 | −224.6 | −214.1 | 4.5 | 4.4 | 5.8 | 7.9 | 0.9 | 17 | 9.43 | 46.99 | 0.99 |
| A | 30 930–43 270 | SOB14-03 | −31.00 | −29.74 | −27.04 | 1.11 | −241.6 | −230.7 | −210.8 | 8.7 | 5.5 | 7.2 | 8.5 | 0.9 | 16 | 7.81 | 1.63 | 0.99 |
| A | 36 970–48 660 | SOB18-09 | −31.38 | −29.66 | −26.85 | 1.08 | −244.3 | −232.0 | −213.1 | 7.8 | 1.7 | 5.2 | 7.5 | 1.1 | 78 | 7.16 | -19.80 | 0.99 |
| A | 49 610 | SOB18-08-I | −30.48 | −29.62 | −28.37 | 0.80 | −237.5 | −230.2 | −218.4 | 6.8 | 5.1 | 6.8 | 8.5 | 1.2 | 10 | 8.28 | 15.08 | 0.97 |









Table 3: Radiocarbon ages of organic material from sediments and ice wedges of the Sobo-Sise Yedoma cliff. n/a stands for not analysed if ages were infinite or beyond the calibration limits (Reimer et al., 2013). BACON-modelled median ages are given for comparison (Fig. S2).


| Sample ID | Height [m arl] | Lab ID | Radiocarbon age [BP] | Material | Calibrated age 2σ range [cal BP] | Calibrated median age [cal BP] | Modelled median age [cal BP] |
|---|---|---|---|---|---|---|---|
| **SEDIMENTS** | | | | | | | |
| SOB18-01-02 | 23.7 | AWI2508.1.1 | 2 389 ± 49 | Cyperaceae remains, moss remains, wood | 2 334 – 2 700 | 2 440 | 2 498 |
| SOB18-01-03 | 23.2 | AWI2509.1.1 | 3 974 ± 49 | Cyperaceae remains, wood | 4 257 – 4 568 | 4 440 | 4 440 |
| SOB18-01-04 | 22.7 | AWI2510.1.1 | 5 597 ± 50 | wood | 6 295 – 6 472 | 6 370 | 6 360 |
| SOB18-01-05 | 22.2 | AWI3921.1.1 | 13 096 ± 71 | twig | 15 389 – 15 970 | 15 710 | 15 483 |
| SOB18-01-06 | 21.7 | AWI3922.1.1 | 13 291 ± 33 | wood | 15 804 – 16 147 | 15 985 | 16 028 |
| SOB18-01-07 | 21.2 | AWI2511.1.1 | 13 841 ± 56 | dwarf-shrub leaf, wood | 16 499 – 16 996 | 16 760 | 16 793 |
| SOB18-01-08 | 20.7 | AWI3923.1.1 | 17 141 ± 113 | twig | 20 367 – 20 996 | 20 680 | 20 419 |
| SOB18-01-09 | 20.2 | AWI3924.1.1 | 17 219 ± 114 | twig | 20 469 – 21 096 | 20 770 | 20 973 |
| SOB18-01-10 | 19.7 | AWI2512.1.1 | 18 102 ± 64 | wood | 21 719 – 22 197 | 21 940 | 21 819 |
| SOB18-01-13 | 18.2 | AWI2513.1.1 | 19 233 ± 68 | vascular plant leaf, wood | 22 928 – 23 446 | 23 170 | 23 357 |
| SOB18-01-15 | 17.2 | AWI2514.1.1 | 20 767 ± 71 | wood | 24 669 – 25 329 | 25 070 | 24 897 |
| SOB18-01-18 | 15.7 | AWI2515.1.1 | 23 305 ± 79 | graminae roots and leaves, wood | 27 369 – 27 716 | 27 540 | 27 555 |
| SOB18-03-02 | 17.7 | AWI2516.1.1 | 21 326 ± 73 | wood | 25 481 – 25 856 | 25 680 | 25 702 |
| SOB18-03-03 | 17.2 | AWI3927.1.1 | 24 408 ± 76 | twigs, fine roots | 28 213 – 28 705 | 28 470 | 28 406 |
| SOB18-03-04 | 16.7 | AWI2517.1.1 | 32 935 ± 117 | Cyperaceae roots and leaves | 35 888 – 36 506 | 36 220 | 36 710 |
| SOB18-03-07 | 15.2 | AWI2518.1.1 | 33 780 ± 216 | Cyperaceae leaves, wood | 37 494 – 38 767 | 38 260 | 38 119 |
| SOB18-03-10 | 13.7 | AWI2519.1.1 | 34 782 ± 236 | Cyperaceae roots, wood | 38 717 – 39 874 | 39 290 | 39 269 |
| SOB18-03-13 | 12.2 | AWI2520.1.1 | 35 965 ± 156 | Cyperaceae stems and roots , 1 *Carex* seed | 40 158 – 41 074 | 40 610 | 40 403 |
| SOB18-03-17 | 10.2 | AWI3928.1.1 | 36 169 ± 121 | Cyperaceae roots and leaves | 40 411 – 41 227 | 40 840 | 41 529 |
| SOB18-03-17 | 10.2 | AWI2521.1.1 | 15 294 ± 67 | Cyperaceae roots, 1 Asteraceae seed, wood | 18 388 – 18 731 | 18 570 | n/a |
| SOB18-06-01 | 13.4 | AWI2522.1.1 | 36 820 ± 302 | *Drepanocladus* stems and leaves | 40 803 – 41 943 | 41 420 | 41 449 |
| SOB18-06-05 | 11.5 | AWI2523.1.1 | 39 877 ± 421 | Cyperaceae roots and stems, *Drepanocladus* stems and leaves | 42 839 – 44 333 | 43 530 | 43 306 |
| SOB18-06-07 | 10.5 | AWI2524.1.1 | 40 572 ± 316 | wood | 43 435 – 44 768 | 44 130 | 44 079 |
| SOB18-06-10 | 9 | AWI2525.1.1 | 43 042 ± 1 726 | Cyperaceae roots and leaves | 43 563 – 49 664 | 46 440 | 45 194 |
| SOB18-06-13 | 7.5 | AWI2526.1.1 | 43 371 ± 431 | Cyperaceae roots and leaves, wood | 45 650 – 47 586 | 46 540 | 46 223 |
| SOB18-06-15 | 6.5 | AWI2527.1.1 | 42 931 ± 414 | Cyperaceae stems, *Drepanocladus* stems and leaves | 45 322 – 47 028 | 46 120 | 46 832 |
| SOB18-06-18 | 5 | AWI2528.1.1 | >42 600 | dwarf shrub leaf, roots | n/a | n/a | 48 046 |
| SOB18-06-20 | 4 | AWI2529.1.1 | 45 345 ± 534 | wood | 47 552 – 49 976 | 48 740 | 48 872 |
| SOB18-06-30 | 3.2 | AWI2530.1.1 | 45 501 ± 542 | 2 Cyperaceae seeds and stems, wood | 47 736 – [50 000] | 48 900 | 49 555 |
| SOB18-06-33 | 1.9 | AWI2531.1.1 | >42 600 | Cyperaceae roots, unspec. wood | n/a | n/a | 50 872 |
| SOB18-06-35 | 0.9 | AWI2532.1.1 | 47 021 ± 646 | Cyperaceae stems | n/a | n/a | 51 877 |
| SOB18-bone-02 | 0 | AWI2749.1.2 | 13 668 ± 57 | ivory (*Mammuthus primigenius*) | 16 255 – 16 751 | 16 480 | n/a |
| SOB18-08-130 | 9.5 | AWI3934.1.1 | 4017 ± 26 | twig, roots | 4 421 – 4 565 | 4 480 | n/a |



| | | | | | | |
|---|---|---|---|---|---|---|
| SOB18-09-100 | 2 | AWI3942.1.1 | >48 500 | 8 *Potamogeton* seeds | n/a | n/a | n/a |

**ICE WEDGES**

| | | | | | | |
|---|---|---|---|---|---|---|
| SOB18-02-B | 19.7 | AWI3925.1.1 | 19 483 ± 148 | lemming droppings | 23 037 – 23 865 | 23 470 | n/a |
| SOB18-02-C | 19.7 | AWI3926.1.1 | 20 991 ± 46 | lemming droppings | 25 149 – 25 538 | 25 350 | n/a |
| SOB18-08-119 | 9.4 | AWI3933.1.1 | 46 169 ± 246 | lemming droppings | 48 981 – [50 000] | 49 610 | n/a |
| SOB18-09-11/12 | 2 | AWI3935.1.1 | 32 963 ± 111 | Poales remains, 5 Poaceae seeds, 1 *Potamogeton* seed, twigs fragments, moss stems and leaves | 36 494 – 37 588 | 36 970 | n/a |
| SOB18-09-29 | 2 | AWI3936.1.1 | 34 577 ± 103 | Poales remains, dwarf shrub leaf fragments, moss stem | 38 698 – 39 453 | 39 040 | n/a |
| SOB18-09-36 | 2 | AWI3937.1.1 | 45 386 ± 842 | lemming droppings | 47 047 – [50 000] | 48 660 | n/a |
| SOB18-09-59 | 2 | AWI3938.1.1 | 42 315 ± 173 | Poales remains, 1 Poaceae seed, dwarf shrub leaf, moss stems and leaves | 45 153 – 46 037 | 45 590 | n/a |
| SOB18-09-63 | 2 | AWI3939.1.1 | 46 601 ± 258 | Cyperaceae leaf, twigs | n/a | n/a | n/a |
| SOB18-09-67 | 2 | AWI3940.1.1 | 43 792 ± 247 | lemming droppings | 46 248 – 47 723 | 46 950 | n/a |
| SOB18-09-79 | 2 | AWI3941.1.1 | 43 437 ± 252 | Poaceae leaves, 5 Poaceae seeds, 2 Fabaceae seeds, 1 Ericaceae seed, moss stems and leaves | 45 933 – 47 294 | 46 570 | n/a |
| SOB14-IW3-02 | 2.5 | AWI1331.1.1 | >31 000 | unidentified organic remains | n/a | n/a | n/a |
| SOB14-IW3-DOC-DAT | 2.5 | COL3809.1.1 | 39 574 ± 374 | unidentified organic remains | 44 041 – 42 696 | 43 270 | n/a |
| SOB14-IW3-09 | 2.5 | AWI1332.1.1 | 26 891 ± 1 025 | unidentified organic remains | 28 818 – 33 202 | 30 930 | n/a |
| SOB14-IW3-15 | 2.5 | AWI1333.1.1 | >31 000 | unidentified organic remains | n/a | n/a | n/a |
| SOB14-IW4-4/12 | 10 | COL3818.1.1 | modern | unidentified organic remains | n/a | n/a | n/a |
| SOB14-IW4-7/01 | 10 | AWI1326.1.1 | 2 264 ± 118 | unidentified organic remains | 1 991 – 2 701 | 2 270 | n/a |
| SOB14-IW4-7/08 | 10 | COL3819.1.1 | 1 617 ± 92 | unidentified organic remains | 1 325 – 1 713 | 1 510 | n/a |
| SOB14-IW4-8/02 | 10 | AWI1327.1.1 | 1 500 ± 115 | unidentified organic remains | 1 182 – 1 692 | 1 410 | n/a |
| SOB14-IW4-8/10 | 10 | AWI1328.1.1 | 1 693 ± 116 | unidentified organic remains | 1 362 – 1 866 | 1 610 | n/a |
| SOB14-IW4-8/10 | 10 | AWI1328.1.2 | 1 491 ± 106 | unidentified organic remains | 1 184 – 1 615 | 1 400 | n/a |
| SOB14-IW4-8/11 | 10 | AWI1329.1.1 | 2 274 ± 119 | unidentified organic remains | 1 996 – 2 703 | 2 290 | n/a |
| SOB14-IW4-9/07 | 10 | AWI1330.1.1 | 2 234 ± 118 | unidentified organic remains | 1 927 – 2 696 | 2 230 | n/a |





**Figures**

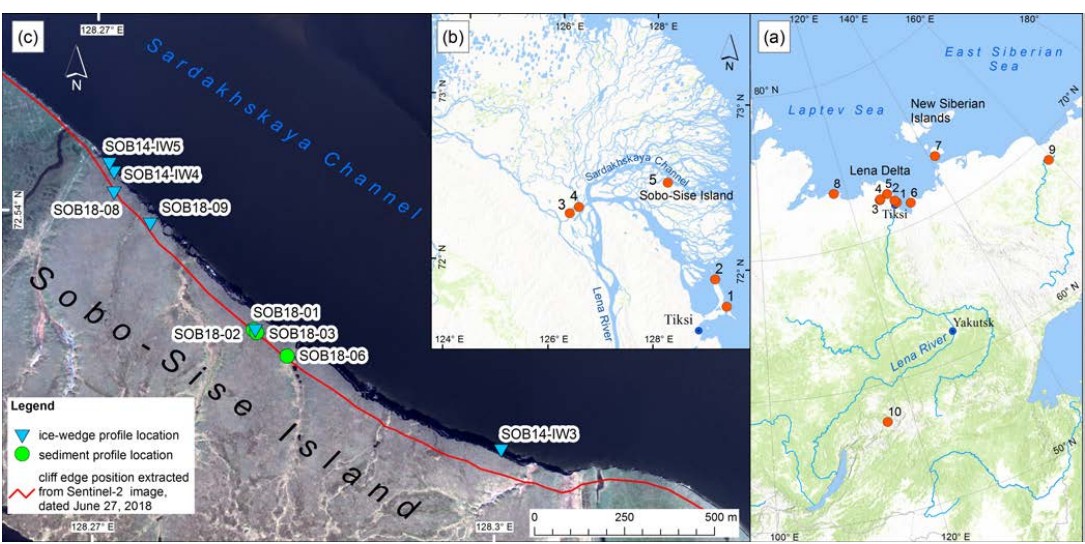

**Figure 1: Study area (a) in north-eastern Siberia showing the study site on Sobo-Sise Island in the eastern Lena Delta (red dot no.**
**5) and further locations mentioned in the paper: Bykovsky Peninsula (1 - site Mamontovy Khayata and  2 - site B-S),  Lena Delta**
**(3 - Kurungnakh-Sise Island, 4 - Samoylov Island), Buor Khaya Peninsula (6), Bol'shoy Lyakhovsky Island (7), Mamontov Klyk**
**(8), Duvanny Yar (9) and Lake Vitim (10). More details of the eastern Lena Delta and the Yedoma IC sites studied nearby Sobo-**
**Sise Island are shown in panel (b). Profile locations are indicated in (c) onsite the Yedoma IC cliff on Sobo-Sise Island (image**
**based on GeoEye-1 scene dated 08 July 2014). Figure 1 (a) and (b) are based on ESRI ArcGIS Living Atlas of the World, layer**
**World Topo Base (2020).**

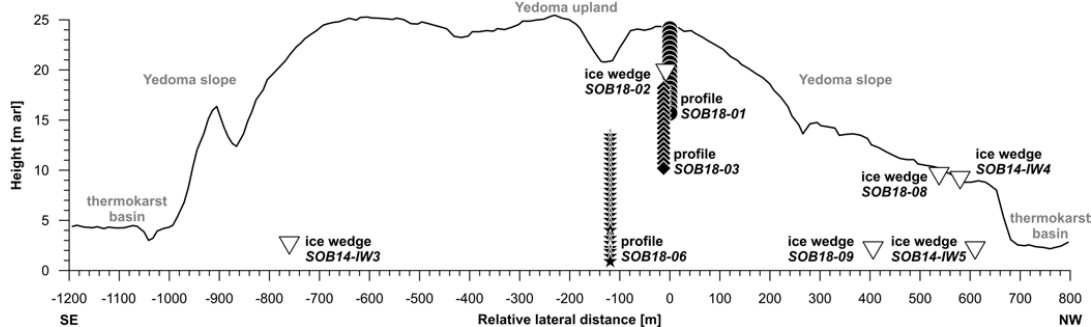

**Figure 2: Profile positions of sediment profiles SOB18-01 (circles), SOB18-03 (diamonds) and SOB18-06 (stars) as well as ice-wedge profiles (triangles) sampled at the Sobo-Sise Yedoma cliff in 2014 (SOB14-...) and 2018 (SOB18-...). Cliff edge line elevation (relative to river level) was retrieved from ArcticDEM (10 m spatial resolution, Mosaic v3.0 10m: tile 59_43; Porter et al., 2018).**






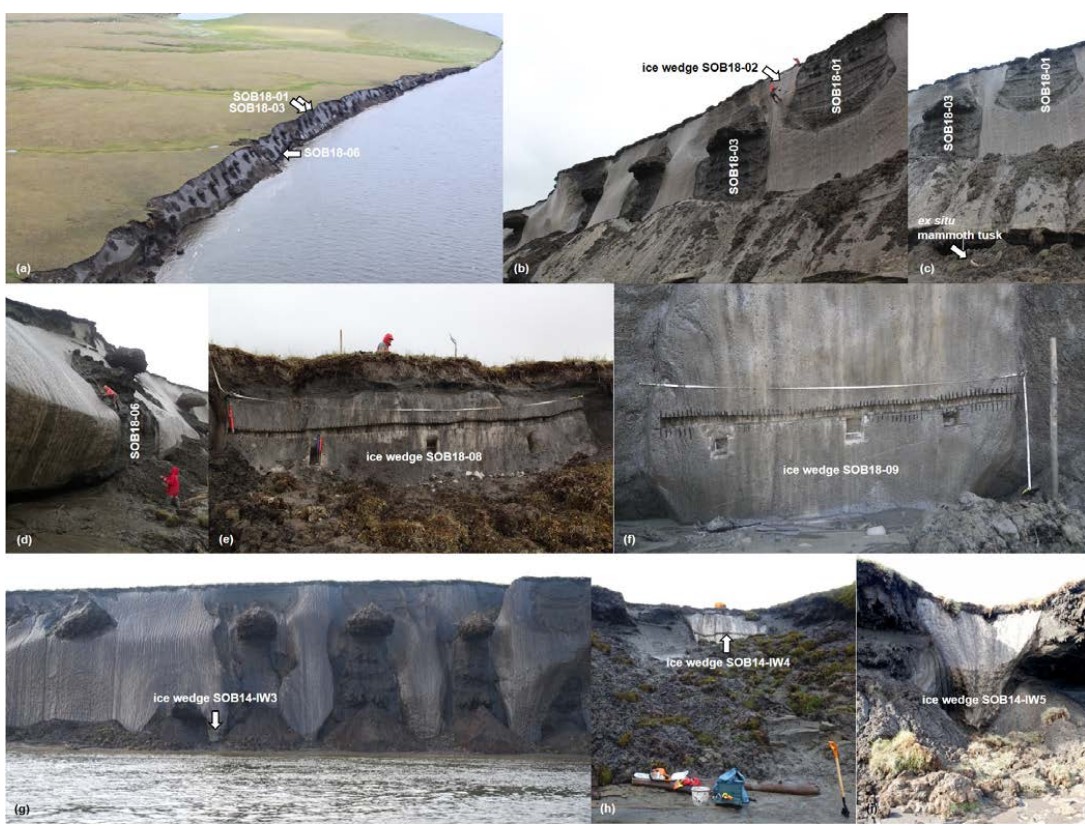

**Figure 3: Field photographs of the sampling locations of the sediment profiles (a-d) and the ice-wedge profiles (b, e-i) at the Sobo-Sise Yedoma cliff. Photographs are provided by the authors of this study.**

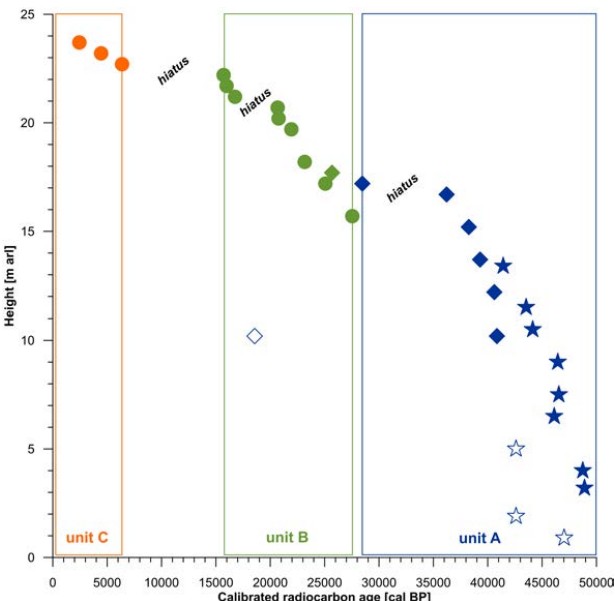

**Figure 4: Age-height relation of the Sobo-Sise Yedoma cliff exposure shown in calibrated radiocarbon ages. Note the sampling overlap of the profiles SOB18-01 (circles), SOB18-03 (diamonds) and SOB18-06 (stars) and their alignment to cryostratigraphic units A (blue), B (green) and C (orange). The open diamond indicates one age of redeposited material from sample SOB18-03-17 and open stars indicate infinite radiocarbon ages of samples from profile SOB18-06.**




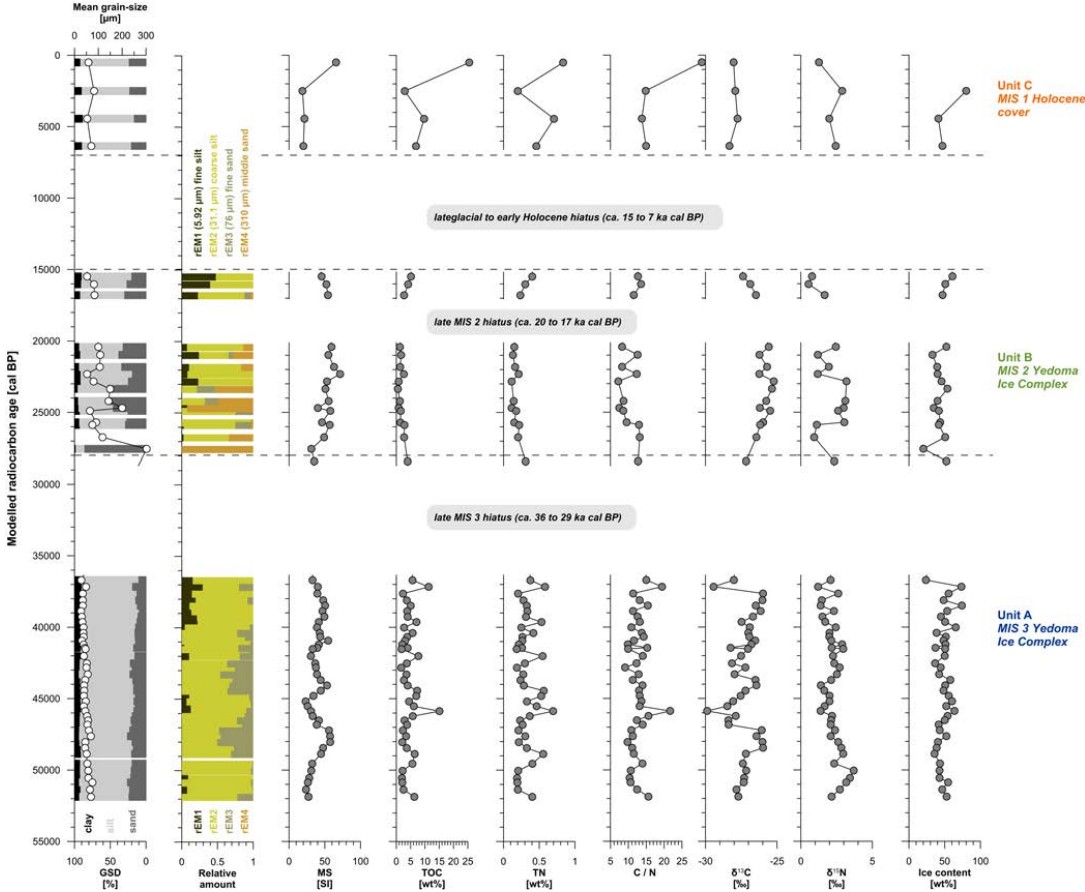

**Figure 5: Sediment properties of the Sobo-Sise Yedoma record and their variations over time. Dashed horizontal lines indicate the**

**limits of the cryostratigraphic units A, B and C. White circles in the plot of grain-size properties relate to the upper x-axis (Mean grain-size).**





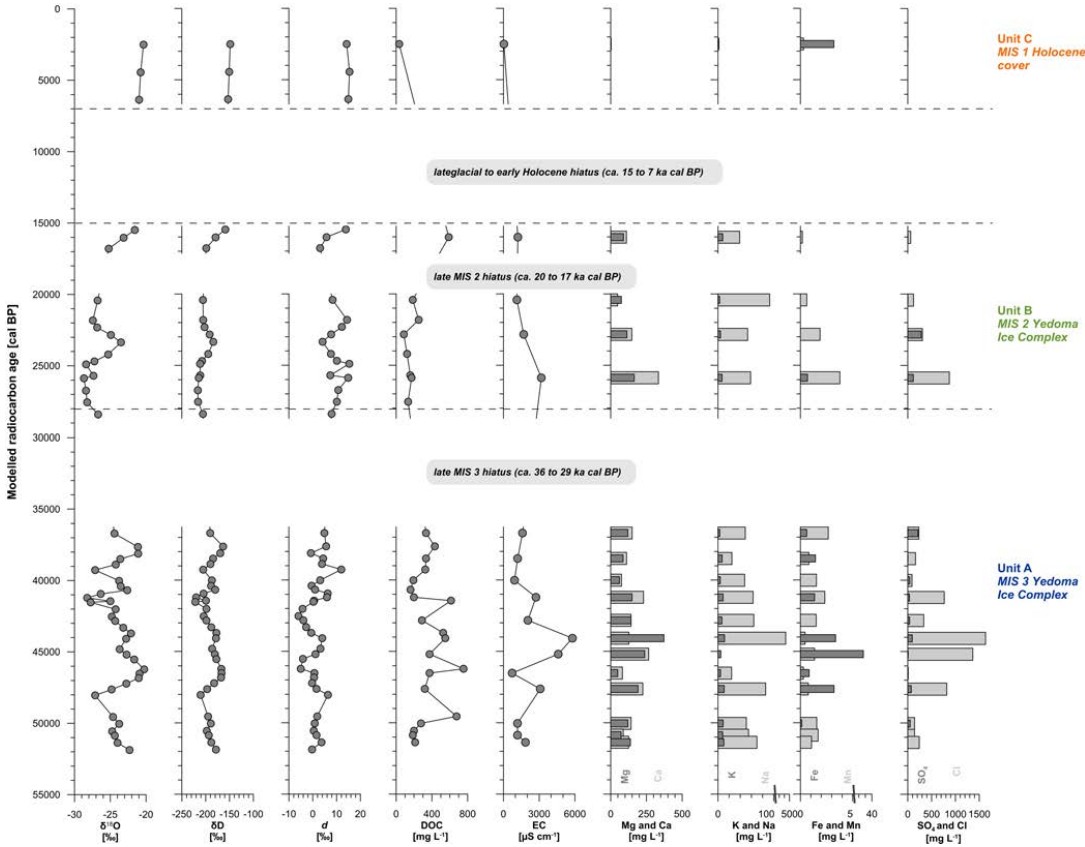

**Figure 6: Intrasedimental ice properties of the Sobo-Sise Yedoma record and their variations over time. Dashed horizontal lines indicate the limits of the cryostratigraphic units A, B and C.**


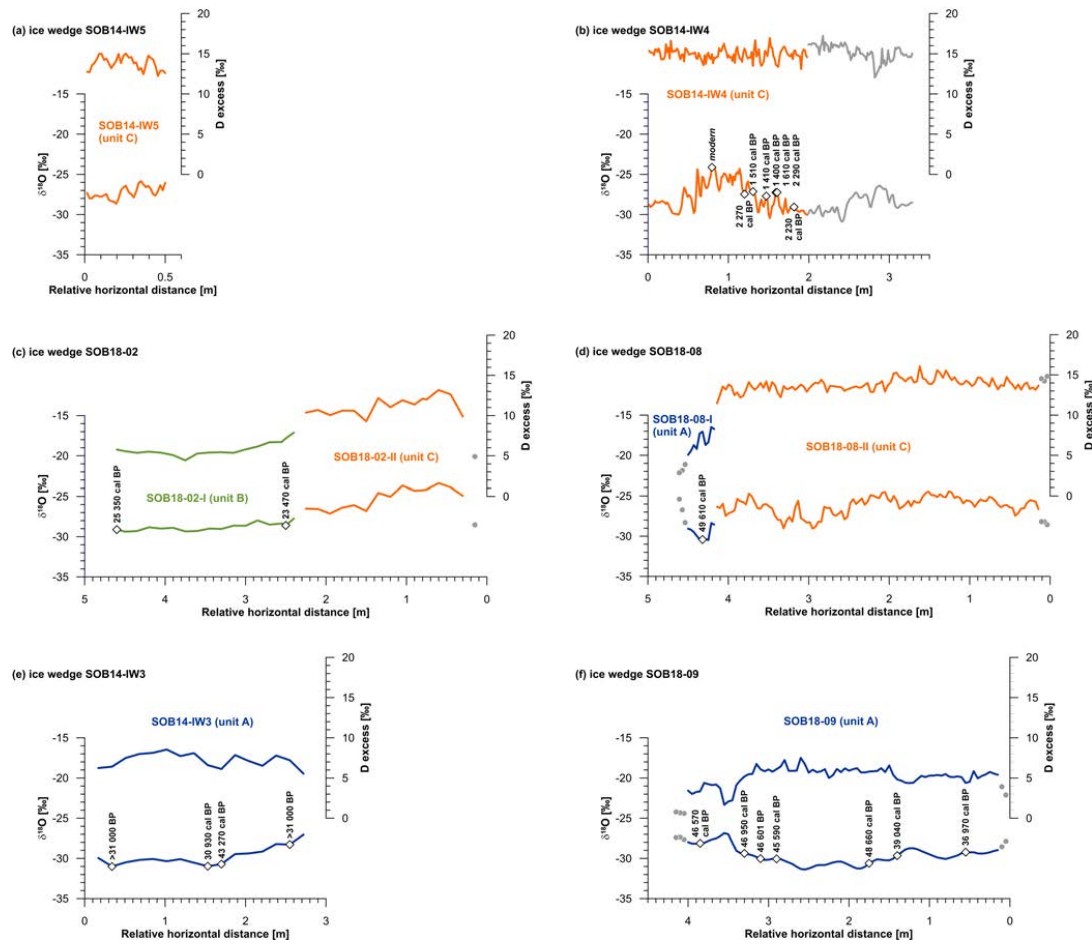

Figure 7: Horizontal wedge ice profiles of the Sobo-Sise Yedoma cliff and their alignment to the cryostratigraphic units A (blue graphs), B (green graphs) and C (orange graphs). Upper graphs refer to the deuterium excess data and the respective right y-axis. Please, note that data points shown in grey are excluded from summary statistics in Table 2. Radiocarbon dates are shown as hollow diamonds and refer to Table 3.





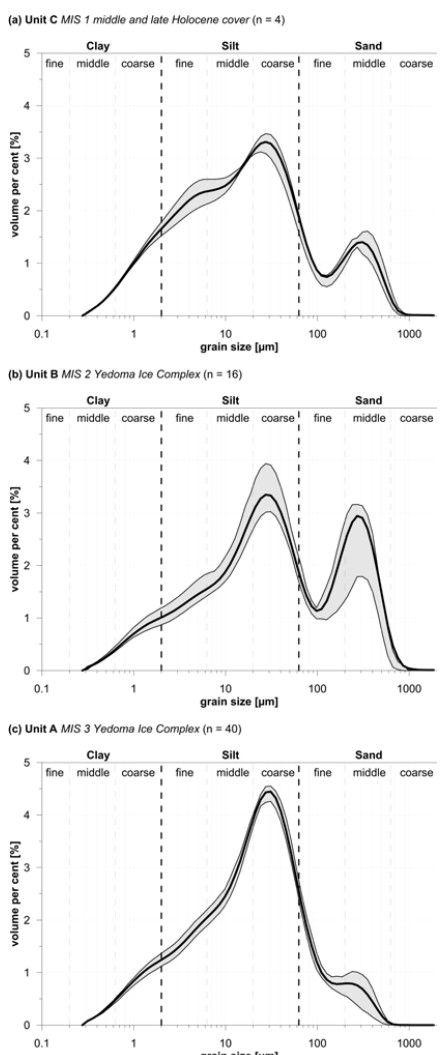

**Figure 8: Grain-size distribution curves for (a) Holocene unit C, (b) MIS 2 unit B and (c) MIS 3 unit A of the Sobo-Sise Yedoma cliff. Bold lines indicate the mean value and grey shaded areas indicate the 25% to 75% quartile.**





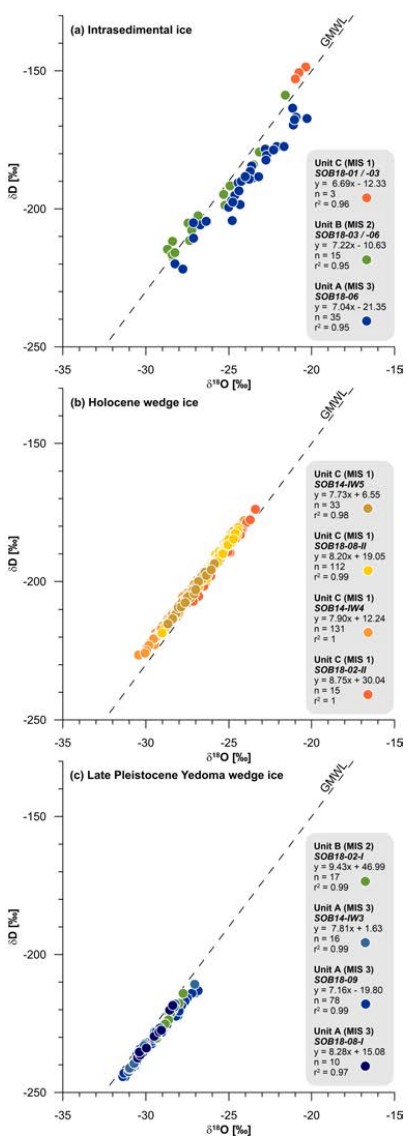

**Figure 9: Stable water isotope composition (δ$^{18}$O, δD) of (a) intrasedimental (excess and pore) ice from units A, B and C, of (b) Holocene wedge ice of unit C and of (c) late Pleistocene Yedoma wedge ice of units B and A of the Sobo-Sise cliff.**



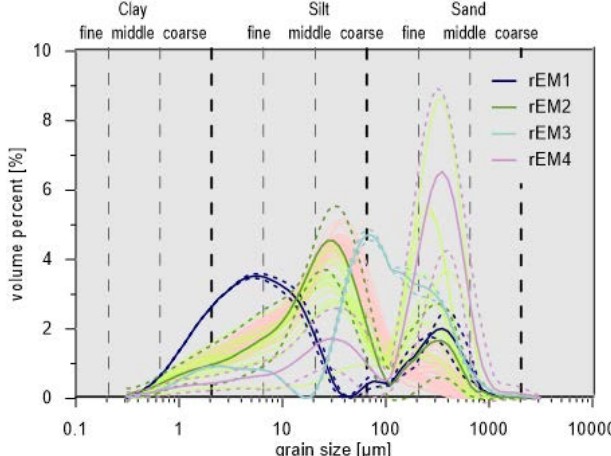

**Figure 10: Grain-size distribution curves and end-member modelling (EMMA) of both Yedoma IC units A and B from the Sobo-Sise Yedoma cliff. EMMA revealed four robust endmembers (rEMs), rEM1 has its primary mode at 5.91 µm in the fine silt, rEM2 has its primary mode at 31.1 µm in the coarse silt. The rEMs 3 and 4 have their primary modes in the fine sand (76 µm) and middle sand (310 µm) respectively.**



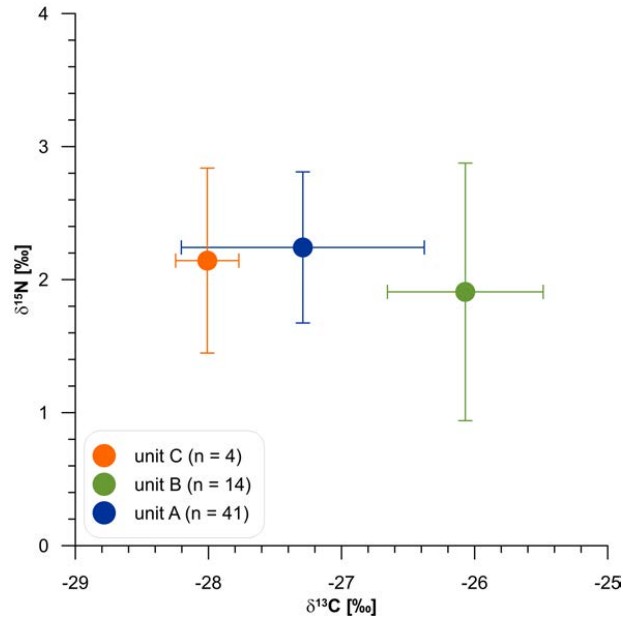


**Figure 11: Stable carbon and nitrogen isotopic composition of organic matter from cryostratigraphic units A, B and C of the Sobo-Sise Yedoma cliff.**
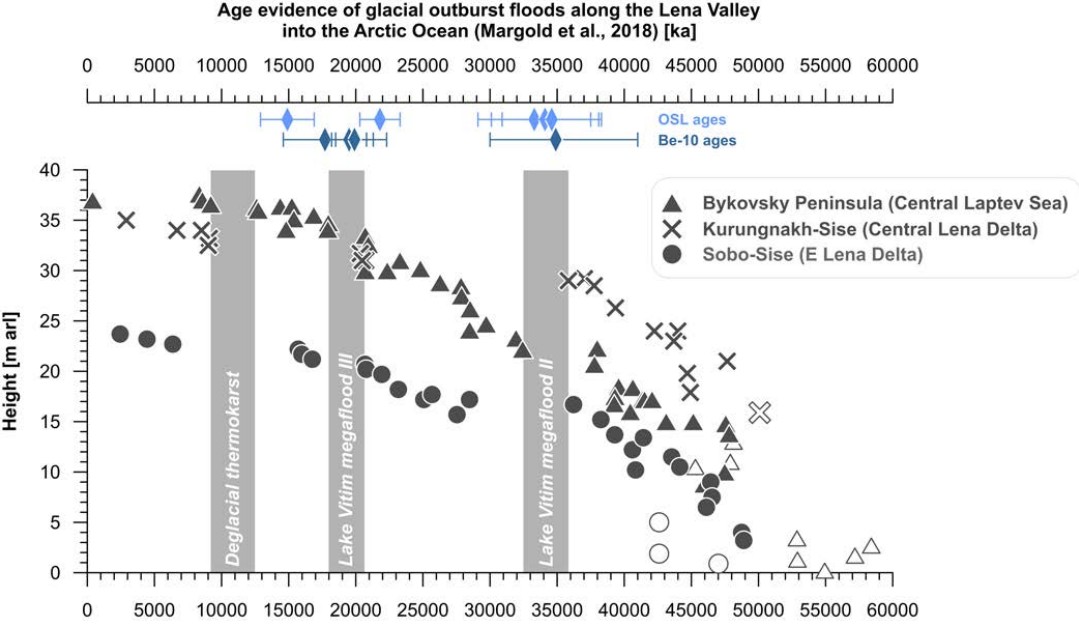

**Figure 12: Comparison of interpreted chronology gaps (shown as shaded rectangles) in the Yedoma IC records from Bykovsky**
**Peninsula (Mamontovy Khayata; Schirrmeister et al., 2002a, 2011), Sobo-Sise Island (this study) and Kurungnakh-Sise Island (Schirrmeister et al., 2003; Wetterich et al., 2008a). Infinite radiocarbon dates or dates to be calibrated beyond the limit of 50 cal ka BP (Reimer et al., 2013) are minimum ages and given as hollow symbols. Age evidence from OSL and Be-10 dating for repeated megafloods (namely numbers II and III) from the glacial Lake Vitim along the Lena Valley into the Arctic Ocean (Margold et al., 2018) is shown for comparison.**