# Peer review of "The cryostratigraphy of the Yedoma cliff of Sobo-Sise Island (Lena Delta) reveals permafrost dynamics in the Central Laptev Sea coastal region during the last about 52 ka"

_The Cryosphere, 2020_

## Referee Comment (RC1) · Anonymous Referee #1 · 25 Aug 2020

Wetterich et al. present a comprehensive physical and geochemical characterisation of a Yedoma exposure on Sobo-Sise Island in the Lena Delta. Yedoma sections across the Siberian Arctic provide a unique window to look back on the Middle and Late Pleistocene and reconstruct environmental and climatic conditions based on a range of proxy indicators. This study complements previous studies of Yedoma exposures from the same region, but notably this study examines the Sobo-Sise section in very high stratigraphic/temporal detail compared to studies of other exposures. The authors discuss a number of sedimentological, cryological, isotopic and geochemical indicators to

understand the environmental conditions associated with the formation of this deposit during Marine Isotope Stages 3, 2 and 1. Robust end-member mixing models are used to identify different sediment sources, and stable water isotopes of pore ice and wedge ice are used to understand the potential value of these different ice types as proxies for local climate and ice systematics. Largely I agree with the analysis and interpretations. The authors also identified chronological gaps in the record that are missing in other Yedoma sections from other sites, which provides clear evidence of a major change in regional environmental conditions that drive Yedoma accumulation. They speculate that two of these gaps may be related to major glacial-lake discharge events due to modified local drainage systems such that available source material of Yedoma accumulation was effectively impacted. It is an interesting, but still unproven point and the authors are careful to not over-interpret this. The paper itself is well written and clear. The methods and study design are scientifically sound. Overall I was impressed with the quality (and quantity) of results and discussion in this paper.

In my view, this paper is well-suited for publication in the Cryosphere. I only have some minor points that should be addressed prior to acceptance, as follows:

L141 - if available, please indicate somewhere what the elevation of river level is in m.a.s.l. L207 – "...not fully..." it is not clear what is meant here. Please clarify this part. L225 – "ballpark" please avoid this colloquialism. L411 – 'Unit', in reference to specific units, is a proper noun and should be capitalized. L456 – suggest using 'relict permafrost' instead of fossil permafrost L477 - This difference would be enhanced if differences in paleo-seawater were explicitly corrected for since mean ocean water during MIS 2 was enriched in heavy isotopologues compared to MIS 3 L506 - this is confusing since 'exceed' typically means values are 'more positive than'. Please use other words to clarify what is meant. L511 - please clarify what is meant. no need to describe the process in detail, but it should be clear what you are talking about. currently it is not clear. L529 - especially when paleo-seawater is considered, the difference is even less significant. L535-537 – following this sentence. "As such..."[please finish this thought]

L625-626 - perhaps you can go one step further to discount direct erosion on the basis that such an event would likely remove tens of meters of sediment, and you are only missing ∼3.5 m given the mean accumulation rate.

---

## Referee Comment (RC2) · Martin Margold (Referee) · 1 Sep 2020

The manuscript 'The cryostratigraphy of the Yedoma cliff of Sobo-Sise Island (Lena Delta) reveals permafrost dynamics in the Central Laptev Sea coastal region during the last about 52 ka' by Wetterich et al. is a fairly technical piece of work that complements the long-term German-Russian research efforts in the permafrost environments of NE Eurasia.

While the core part of the manuscript is outside of my field of expertise, I was ap-

proached to review the manuscript because the authors adopt the suggestion that megafloods from glacial Lake Vitim could have been responsible for the observed hiatus in the chronostratigraphy of the studied sedimentary records. From my perspective, the discussion of the megafloods causing erosion of the sedimentary record or redirecting the main stream of the river into another portion of the delta is sensible, with alternative explanations also considered. I agree also with the argumentation that the stage 3 hiatus does not match the period of the mildest climate and instead postdates it by at least several thousand years.

At line 614, I would suggest to change the wording of 'but might be related to general MIS 3 climate instability' to 'but instead falls to a period of late MIS 3 climate instability...'. This wording is less speculative and the possible linkage between the IC stratigraphy and the DO events is expressed sufficiently in the following sentence.

I do not have any major comments or objections to the manuscript and find it ready for publication in TC.

Minor comment: Fig. 1. It would be good to include the River Vitim in panel (a), given that the floods that went down the Vitim-Lena route are a key part of the interpretations of the studied sedimentary record.
* * *

---

## Author Comment (AC2) · 22 Sep 2020

Author's response to the interactive comment on "The cryostratigraphy of the Yedoma cliff of Sobo-Sise Island (Lena Delta) reveals permafrost dynamics in the Central Laptev Sea coastal region during the last about 52 ka" by Sebastian Wetterich et al. Martin Margold (Referee) #2, https://doi.org/10.5194/tc-2020-179-RC2

The manuscript 'The cryostratigraphy of the Yedoma cliff of Sobo-Sise Island (Lena

[Figure]

Delta) reveals permafrost dynamics in the Central Laptev Sea coastal region during thelast about 52 ka' by Wetterich et al. is a fairly technical piece of work that complements the long-term German-Russian research efforts in the permafrost environments of NE Eurasia. While the core part of the manuscript is outside of my field of expertise, I was approached to review the manuscript because the authors adopt the suggestion that megafloods from glacial Lake Vitim could have been responsible for the observed hiatus in the chronostratigraphy of the studied sedimentary records. From my perspective, the discussion of the megafloods causing erosion of the sedimentary record or redi-recting the main stream of the river into another portion of the delta is sensible, with alternative explanations also considered. I agree also with the argumentation that the stage 3 hiatus does not match the period of the mildest climate and instead postdates it by at least several thousand years.

REPLY - Thank you for your positive statement and your approval relating the Margold et al. (2018) study to our interpretation of the hiatuses observed in the Yedoma Ice Complex records on Sobo Sise, Kurungnakh-Sise island of the Lena Delta as well as on Bykovsky Peninsula.

At line 614, I would suggest to change the wording of 'but might be related to general MIS 3 climate instability' to 'but instead falls to a period of late MIS 3 climate instability...'. This wording is less speculative and the possible linkage between the IC stratigraphy and the DO events is expressed sufficiently in the following sentence.

REPLY - Changed accordingly.

I do not have any major comments or objections to the manuscript and find it ready for publication in TC. Minor comment

Fig. 1. It would be good to include the River Vitim in panel (a), given that the floods that went down the Vitim-Lena route are a key part of the interpretations of the studied sedimentary record.

REPLY - Changed accordingly.

Figure 1: Study area (a) in north-eastern Siberia showing the study site on Sobo-Sise Island in the eastern Lena Delta (red dot no. 5) and further locations mentioned in the paper: Bykovsky Peninsula (1 - site Mamontovy Khayata and 2 - site B-S), Lena Delta (3 - Kurungnakh-Sise Island, 4 - Samoylov Island), Buor Khaya Peninsula (6), Bol'shoy Lyakhovsky Island (7), Mamontov Klyk (8), Duvanny Yar (9) and Lake Vitim (10). More details of the eastern Lena Delta and the Yedoma IC sites studied nearby Sobo-Sise Island are shown in panel (b). Profile locations are indicated in (c) onsite the Yedoma IC cliff on Sobo-Sise Island (image based on GeoEye-1 scene dated 08 July 2014). Figure 1 (a) and (b) are based on ESRI ArcGIS Living Atlas of the World, layer World Topo Base (2020).
* * *
[Figure]

[Figure]

**Fig. 1.** Updated Figure 1

---

## Author Response (AR1)

Dear Dr. Sjöberg,

thank you for your overall thoughtful reception of our submission. We further appreciate the work of both reviewers commenting on our paper. Our replies (blue letters) are given in detail below.

We further proofread the entire ms and corrected typos in the text. We updated Figures 8 and 10 showing grain-size distributions and endmember modelling results to make both graphical representations more easily comparable. In Figure 10, we changed the colour code of rEMs; now corresponding with that of Figure 5.

On behalf of the authors,

Sebastian Wetterich

**Editor's comments to the Author**

The Cryosphere is committed to the FAIR ("findable, accessible, interoperable, and reusable") research principles (https://www.the-cryosphere.net/about/data_policy.html). Following this, before publication, the authors need to add a data availability section, even if all data is presented in the manuscript.

**REPLY:** We added the reference of the original data published in PANGAEA to section Data availability and to the reference list: "*Wetterich, S., Meyer, H., Fritz, M., Opel, T., Schirrmeister, L.: Cryolithology of the Sobo-Sise Yedoma cliff (eastern Lena Delta), PANGAEA, https://doi.org/10.1594/PANGAEA.919470, 2020.*"

Before publication the cited publications which are currently under review (Fuchs et al., cited heavily) should be accepted or published.

**REPLY:** We updated the reference list accordingly: Fuchs, M., Nitze, I., Strauss, J., Günther, F., Wetterich, S., Kizyakov, A., Fritz, M., Opel, T., Grigoriev, M. N., Maximov, G.M., and Grosse, G.: Rapid fluvio-thermal erosion of a yedoma permafrost cliff in the Lena River Delta, Front. Earth Sci., 8, 336, https://doi.org/10.3389/feart.2020.00336, 2020.

We further deleted the reference to Murton et al. that is still in review.: Murton, J., Opel, T., Toms, P., Blinov, A., Fuchs, M., Woods, J., Gärtner, A., Merchel, S., Rugel, G., Savvinov, G. and Wetterich, S.: A multi-method pilot dating study of ancient permafrost, Batagay megaslump, East Siberia, Quaternary Res., under review.

Author's response to the interactive comment on "The cryostratigraphy of the Yedoma cliff of Sobo-Sise Island (Lena Delta) reveals permafrost dynamics in the Central Laptev Sea coastal region during the last about 52 ka" by Sebastian Wetterich et al.

**Anonymous Referee #1, https://doi.org/10.5194/tc-2020-179-RC1**

Wetterich et al. present a comprehensive physical and geochemical characterisation of a Yedoma exposure on Sobo-Sise Island in the Lena Delta. Yedoma sections across the Siberian Arctic provide a unique window to look back on the Middle and Late Pleistocene and reconstruct environmental and climatic conditions based on a range of proxy indicators. This study complements previous studies of Yedoma exposures from the same region, but notably this study examines the Sobo-Sise section in very high stratigraphic/temporal detail compared to studies of other exposures. The authors discuss a number of sedimentological, cryological, isotopic and geochemical indicators to understand the environmental conditions associated with the formation of this deposit during Marine Isotope Stages 3, 2 and 1. Robust end-member mixing models are used to identify different sediment sources, and stable water isotopes of pore ice and wedge ice are used to understand the potential value of these different ice types as proxies for local climate and ice systematics. Largely I agree with the analysis and interpretations. The authors also identified chronological gaps in the record that are missing in other Yedoma sections from other sites, which provides clear evidence of a major change in regional environmental conditions that drive Yedoma accumulation. They speculate that two of these gaps may be related to major glacial-lake discharge events due to modified local drainage systems such that available source material of Yedoma accumulation was effectively impacted. It is an interesting, but still unproven point and the authors are careful to not over-interpret this. The paper itself is well written and clear. The methods and study design are scientifically sound. Overall I was impressed with the quality (and quantity) of results and discussion in this paper. In my view, this paper is well-suited for publication in the Cryosphere.

     **REPLY:** Thank you for your time and effort to review our manuscript. We appreciate your overall positive feedback on our study.

I only have some minor points that should be addressed prior to acceptance, as follows:

     **REPLY:** Your minor points have been carefully addressed as outlined below.

**L141** – if available, please indicate somewhere what the elevation of river level is in m.a.s.l.

     **REPLY:** Changed accordingly by adding the following statement in section 3.1 "*Height measures in m arl correspond to those above sea level (m asl), given the proximity of Sobo-Sise Island at the Sardakhskaya Channel in the eastern part of the Lena Delta to the Laptev Sea (Fig. 1 b)."*

**L207** – "...not fully...' it is not clear what is meant here. Please clarify this part.

     **REPLY:** We specified the sentence as follows: "*We considered only a full profile of one wedge cut and sampled perpendicular to its lateral growth direction and neglected the remaining samples of the second ice wedge, not completely captured due to its oblique exposition."*

**L225** – "ballpark" please avoid this colloquialism.

     **REPLY:** Changed accordingly to "*approximate age"*.

**L411** – 'Unit', in reference to specific units, is a proper noun and should be capitalized.

     **REPLY:** Changed accordingly throughout the manuscript.

**L456** – suggest using 'relict permafrost' instead of fossil permafrost

     **REPLY:** Changed accordingly.

**L477** – This difference would be enhanced if differences in paleo-seawater were explicitly corrected for since mean ocean water during MIS 2 was enriched in heavy isotopologues compared to MIS 3

**REPLY:** Indeed the difference between MIS 3 and MIS 2 isotopic signatures would be increased when considering the paleo-ocean water enrichment in heavy isotopologues during MIS 2 due to the storage of isotopically light water in ice sheets. We, however did not decide to apply this source-water correction because our wedge-ice stable isotope records from Sobo-Sise are not suitable for quantitative paleo-temperature reconstruction due to the low temporal resolution of the record based on [14]C dates and the yet not satisfying relation between wedge-ice stable isotope composition and winter temperature (Porter & Opel, 2020)

Reference: Porter, T. J., and Opel, T.: Recent advances in paleoclimatological studies of Arctic wedge and pore ice stable water isotope records. Permafrost and Periglac., 31, 429– 441, https://doi.org/10.1002/ppp.2052, 2020.

**L506** – this is confusing since 'exceed' typically means values are 'more positive than'. Please use other words to clarify what is meant.

**REPLY:** Sentence changed accordingly to "In some instances, Holocene $\delta^{18}$O and $\delta$D values reach the range of the late Pleistocene ice wedges (Table 2)."

**L511** – please clarify what is meant. No need to describe the process in detail, but it should be clear what you are talking about. Currently it is not clear.

**REPLY:** We removed the unclear statement on possible isotopic diffusion within wedge ice over time that is beyond the scope of our study. Deleted sentences: "It is obvious that the horizontal profiles of late Pleistocene ice wedges are less spiky than those of Holocene ice wedges (Fig. 7). This might indicate a time-dependent smoothing of the isotope profiles due to isotopic diffusion within ice wedges but is beyond the scope of this study."

**L529** – especially when paleo-seawater is considered, the difference is even less significant.

**REPLY:** Agreed. See our reply to your comment on ln477.

**L535-537** – following this sentence. "As such..." [please finish this thought]

**REPLY:** To clarify the statement, we changed the order of sentences as follows: "*This might indicate that the globally cold LGM is not reflected in the Sobo-Sise ice wedge-based winter climate record and would be in accordance with both regional scale, when compared to Bykovsky Peninsula (Meyer et al., 2002a) or to other study sites in the Laptev Sea region (Wetterich et al., 2011), and also on Arctic-wide scale (Porter and Opel, 2020). In this context, we observe (1) a depositional gap temporally coinciding to peak LGM conditions for the three sites at regional scale and (2) extremely depleted LGM ice-wedge isotopes have been only found at Bol'shoy Lyakhovsky Island further east (Fig. 1; Wetterich, et al., 2011). As such it is not sufficiently resolved yet, whether this is due to a less cold LGM climate in the region or whether the LGM cold period is not captured by the studied ice-wedge profiles that do not preserve a continuous record.*"

**L625-626** – perhaps you can go one step further to discount direct erosion on the basis that such an event would likely remove tens of meters of sediment, and you are only missing~3.5 m given the mean accumulation rate.

**REPLY:** Agreed. We added the following statement to the sentence: "But except for the chronology gaps no direct erosional features such as fluvial sand or pebble layers have been observed in the outcrops. Thus, direct erosion seems unlikely at the studied locations and the flooding events may here have only changed the hydrological regime (e.g. by developing new discharge paths similar to the channels in the today's Lena Delta) for a certain time period and by doing so prevented the deposition of fluvially transported material in the areas of the studied Yedoma IC outcrops."

Author's response to the interactive comment on "The cryostratigraphy of the Yedoma cliff of Sobo-Sise Island (Lena Delta) reveals permafrost dynamics in the Central Laptev Sea coastal region during the last about 52 ka" by Sebastian Wetterich et al.

**Martin Margold (Referee) #2, https://doi.org/10.5194/tc-2020-179-RC2**

The manuscript 'The cryostratigraphy of the Yedoma cliff of Sobo-Sise Island (Lena Delta) reveals permafrost dynamics in the Central Laptev Sea coastal region during the last about 52 ka' by Wetterich et al. is a fairly technical piece of work that complements the long-term German-Russian research efforts in the permafrost environments of NE Eurasia. While the core part of the manuscript is outside of my field of expertise, I was approached to review the manuscript because the authors adopt the suggestion that megafloods from glacial Lake Vitim could have been responsible for the observed hiatus in the chronostratigraphy of the studied sedimentary records. From my perspective, the discussion of the megafloods causing erosion of the sedimentary record or redirecting the main stream of the river into another portion of the delta is sensible, with alternative explanations also considered. I agree also with the argumentation that the stage 3 hiatus does not match the period of the mildest climate and instead postdates it by at least several thousand years.

> **REPLY:** Thank you for your positive statement and your approval relating the Margold et al. (2018) study to our interpretation of the hiatuses observed in the Yedoma Ice Complex records on Sobo Sise, Kurungnakh-Sise island of the Lena Delta as well as on Bykovsky Peninsula.

At **line 614**, I would suggest to change the wording of 'but might be related to general MIS 3 climate instability' to 'but instead falls to a period of late MIS 3 climate instability...'. This wording is less speculative and the possible linkage between the IC stratigraphy and the DO events is expressed sufficiently in the following sentence.

> **REPLY:** Changed accordingly.

I do not have any major comments or objections to the manuscript and find it ready for publication in TC.

Minor comment

**Fig. 1.** It would be good to include the River Vitim in panel (a), given that the floods that went down the Vitim-Lena route are a key part of the interpretations of the studied sedimentary record.

> **REPLY:** Changed accordingly.

[revised manuscript text omitted]

Sebastian Wetterich 14.9.20 13:27
Kommentar [5]: rev#1: L456 – suggest using 'relict permafrost' instead of fossil permafrost reply: changed accordingly
Sebastian Wetterich 14.9.20 13:28

compared to MIS 2 (Fig. 6; Fig. 9). Relatively warm summers during the MIS 3 interstadial might explain the lower *d* values in associated intrasedimental ice due to a higher water loss by evaporation (i.e., kinetic fractionation). This would lead to a water reservoir in polygon ponds and soil moisture that becomes successively depleted in $^{16}$O and $^1$H compared to the original precipitation. Increased temperature and precipitation amplitudes during MIS 3 (Andreev et al., 2011; Pitulko et al.,

505 2017) may have led to frequent drying and re-wetting in polygon tundra and thus to enhanced kinetic fractionation. Another process of kinetic fractionation producing the same pattern are multiple freeze-thaw cycles of soil moisture in the active layer (Throckmorton et al., 2016).

Elevated ion (Mg, Ca, Na, Cl) concentrations with EC up to 5800 µScm$^{-1}$ in the MIS 3 record (Unit A; Fig. 6) are likely caused by frequent drying and re-wetting in polygonal tundra in times of higher summer temperature and precipitation

510 amplitudes during the interstadial compared to MIS 2 stadial (Unit B). Meyer et al. (2002a) found similarly elevated EC values of 5500 µScm$^{-1}$ in MIS 3 deposits on Bykovsky Peninsula. Modern surface waters in Central Yakutia at high continentality show EC values of up to 5710 µScm$^{-1}$ (Wetterich et al., 2008b) and even up to 7744 µScm$^{-1}$ (Pestryakova et al., 2018). Ion-rich pore waters have also been found in MIS 3 deposits at Buor Khaya Peninsula (Schirrmeister et al., 2017), but with different composition and including a distinct saline horizon. In contrast, ion composition in the Sobo-Sise Yedoma

515 IC remained stable throughout MIS 2 and MIS 3 and is dominated by Mg, Cl and Ca in both units. Therefore, we assume that water and sediment sources did not change over time, but reflect higher evaporation during warmer summers in MIS 3 if compared to MIS 2.

**5.2 Palaeoclimatic implications from regional wedge-ice records**

Sobo-Sise ice wedge stable isotopes show a complex pattern that at least in parts can be related to the fact that Holocene ice

520 wedges formed epigenetically within older late Pleistocene deposits and penetrated pre-existing ice wedges. This may be related to subsidence and thermo-erosional processes that thaw permafrost, lower the surface and complicate the stratigraphic attribution of the wedge ice. The stable isotope composition of ice wedge profiles sampled in the central (SOB18-02) and western parts (SOB18-08) of the Sobo-Sise cliff allows differentiating late Pleistocene and Holocene wedge ice.

525 Generally, late Pleistocene wedge ice is characterised by well-depleted δ$^{18}$O and δD values (mean values between –30 ‰ and –29 ‰, and –232 ‰ and –225 ‰, respectively; Fig. 9 c) and low *d* values (means between 5‰ and 7‰; Table 2). In contrast, a striking feature of Holocene ice wedges are their significantly elevated mean *d* values between 11‰ and 15‰ (Table 2) accompanied by surprisingly low δ$^{18}$O and δD values (mean values between –28 ‰ and –25 ‰, and –207 ‰ and – 190 ‰, respectively; Fig. 9 b). In some instances, Holocene δ$^{18}$O and δD values reach the range of the late Pleistocene ice

530 wedges (Table 2). This is true for both the oldest Holocene ice wedge stage, i.e. the toes of ice wedge SOB14-IW5 at the Ice Complex-Alas slope and the late Holocene to modern ice wedges on the top of the Ice Complex (e.g. SOB14-IW4). Hence, the isotopic difference between late Pleistocene and Holocene ice wedges is more pronounced in *d* than in δ values.

Sebastian Wetterich 9.9.20 09:49
**Kommentar [6]:** rev#1: L477 – This difference would be enhanced if differences in paleo-seawater were explicitly corrected for since mean ocean water during MIS 2 was enriched in heavy isotopologues compared to MIS 3
reply: This difference would be more pronounced when considering the isotopically more enriched palaeo-ocean water (about 0.6 o/oo in d18O and 5 o/oo in dD, porter et al. 2016, pp.118 oben) during MIS 2 if compared to MIS 3.

Sebastian Wetterich 14.9.20 13:41

Sebastian Wetterich 14.9.20 15:26
**Kommentar [7]:** rev#1: L506 – this is confusing since 'exceed' typically means values are 'more positive than'. Please use other words to clarify what is meant.

Sebastian Wetterich 14.9.20 15:24
**Kommentar [8]:** rev#1: L511 – please clarify what is meant. no need to describe the process in detail, but it should be clear what you are talking about. currently it is not clear.

Sebastian Wetterich 17.9.20 09:47

[revised manuscript text omitted]

Sebastian Wetterich 4.9.20 10:36

**Kommentar [15]:** rev#2: **Fig. 1.** It would be good to include the River Vitim in panel (a), given that the floods that went down the Vitim-Lena route are a key part of the interpretations of the studied sedimentary record.

[Figure]

155 **Figure 2: Profile positions of sediment profiles SOB18-01 (circles), SOB18-03 (diamonds) and SOB18-06 (stars) as well as ice-wedge profiles (triangles) sampled at the Sobo-Sise Yedoma cliff in 2014 (SOB14-...) and 2018 (SOB18-...). Cliff edge line elevation (relative to river level) was retrieved from ArcticDEM (10 m spatial resolution, Mosaic v3.0 10m: tile 59_43; Porter et al., 2018).**

[Figure]

160 **Figure 3: Field photographs of the sampling locations of the sediment profiles (a-d) and the ice-wedge profiles (b, e-i) at the Sobo-Sise Yedoma cliff. Photographs are provided by the authors of this study.**

[Figure]

**Figure 4: Age-height relation of the Sobo-Sise Yedoma cliff exposure shown in calibrated radiocarbon ages. Note the sampling overlap of the profiles SOB18-01 (circles), SOB18-03 (diamonds) and SOB18-06 (stars) and their alignment to cryostratigraphic units A (blue), B (green) and C (orange). The open diamond indicates one age of redeposited material from sample SOB18-03-17 and open stars indicate infinite radiocarbon ages of samples from profile SOB18-06.**

1165

[Figure]

**Figure 5: Sediment properties of the Sobo-Sise Yedoma record and their variations over time. Dashed horizontal lines indicate the limits of the cryostratigraphic units A, B and C. White circles in the plot of grain-size properties relate to the upper x-axis (Mean grain-size).**

1170

[Figure]

**Figure 6: Intrasedimental ice properties of the Sobo-Sise Yedoma record and their variations over time. Dashed horizontal lines indicate the limits of the cryostratigraphic units A, B and C.**

[Figure]

**Figure 7: Horizontal wedge ice profiles of the Sobo-Sise Yedoma cliff and their alignment to the cryostratigraphic units A (blue graphs), B (green graphs) and C (orange graphs). Upper graphs refer to the deuterium excess data and the respective right y-axis. Please, note that data points shown in grey are excluded from summary statistics in Table 2. Radiocarbon dates are shown as hollow diamonds and refer to Table 3.**

[Figure]

**Figure 8:** Grain-size distribution curves for (a) Holocene unit C, (b) MIS 2 unit B and (c) MIS 3 unit A of the Sobo-Sise Yedoma cliff. Bold lines indicate the mean value and grey shaded areas indicate the 25% to 75% quartile.

180

Sebastian Wetterich 7.9.20 15:16

**Kommentar [16]:** Updated Figure 8.

[Figure]

**Figure 9: Stable water isotope composition (δ¹⁸O, δD) of (a) intrasedimental (excess and pore) ice from units A, B and C, of (b)**
1185 **Holocene wedge ice of unit C and of (c) late Pleistocene Yedoma wedge ice of units B and A of the Sobo-Sise cliff.**

[Figure]

**Figure 10:** Grain-size distribution curves and end-member modelling (EMMA) of both Yedoma IC units A and B from the Sobo-Sise Yedoma cliff. EMMA revealed four robust endmembers (rEMs), rEM1 has its primary mode at 5.91 µm in the fine silt, rEM2 has its primary mode at 31.1 µm in the coarse silt. The rEMs 3 and 4 have their primary modes in the fine sand (76 µm) and middle sand (310 µm) respectively.

Sebastian Wetterich 7.9.20 15:18
**Kommentar [17]:** Updated Figure 10. Now rEM color code in accordance with Figure 5.

[Figure]

Figure 11: Stable carbon and nitrogen isotopic composition of organic matter from cryostratigraphic units A, B and C of the Sobo-Sise Yedoma cliff.

[Figure]

**Figure 12: Comparison of interpreted chronology gaps (shown as shaded rectangles) in the Yedoma IC records from Bykovsky Peninsula (Mamontovy Khayata; Schirrmeister et al., 2002a, 2011), Sobo-Sise Island (this study) and Kurungnakh-Sise Island (Schirrmeister et al., 2003; Wetterich et al., 2008a). Infinite radiocarbon dates or dates to be calibrated beyond the limit of 50 cal ka BP (Reimer et al., 2013) are minimum ages and given as hollow symbols. Age evidence from OSL and Be-10 dating for repeated megafloods (namely numbers II and III) from the glacial Lake Vitim along the Lena Valley into the Arctic Ocean (Margold et al., 2018) is shown for comparison.**

1200